



# HTAP3 Fires: Towards a multi-model, multi-pollutant study of fire impacts

Cynthia H. Whaley[1], Tim Butler[2], Jose A. Adame[3], Rupal Ambulkar[4,5], Steve R. Arnold[6], Rebecca R. Buchholz[7], Benjamin Gaubert[7], Douglas S. Hamilton[8], Min Huang[9], Hayley Hung[10], Johannes W. Kaiser[11], Jacek W. Kaminski[12], Christoph Knote[13], Gerbrand Koren[14], Jean-Luc Kouassi[15], Meiyun Lin[16], Tianjia Liu[17], Jianmin Ma[18], Kasemsan Manomaiphiboon[19], Elisa Bergas Masso[20,21], Jessica L. McCarty[22], Mariano Mertens[23], Mark Parrington[24], Helene Peiro[25], Pallavi Saxena[26], Saurabh Sonwani[27], Vanisa Surapipith[28], Damaris Tan[29,30], Wenfu Tang[7], Veerachai Tanpipat[31], Kostas Tsigaridis[32,33], Christine Wiedinmyer[34], Oliver Wild[35], Yuanyu Xie[36], Paquita Zuidema[37]

[1]Climate Research Division, Environment and Climate Change Canada, Victoria, BC, Canada
[2]Research Institute for Sustainability – Helmholtz Centre Potsdam, Germany
[3]Atmospheric Sounding Station, El Arenosillo, National Institute for Aerospace Technology (INTA), Mazagón-Huelva. Spain
[4]Indian Institute of Tropical Meteorology (IITM), Pune, India
[5]Department of Environmental Sciences, Savitribai Phule Pune University, Pune, India
[6]Institute for Climate and Atmospheric Science, School of Earth and Environment, University of Leeds, UK.
[7]Atmospheric Chemistry Observations & Modelling, National Science Foundation (NSF) National Center for Atmospheric Research (NCAR), Boulder, Colorado, 80501, USA
[8] Marine, Earth, and Atmospheric Sciences, North Carolina State University, Raleigh, NC, USA
[9]Earth System Science Interdisciplinary Center, University of Maryland, College Park, MD, USA
[10]Air Quality Processes Research Section, Environment and Climate Change Canada, Toronto, ON, Canada.
[11]NILU, Department for Atmospheric and Climate Research, Kjeller, Norway
[12]Institute of Environmental Protection - National Research Institute, Warsaw, Poland
[13]Model-based Environmental Exposure Science, Faculty of Medicine, University of Augsburg, Germany
[14]Copernicus Institute of Sustainable Development, Utrecht University, Utrecht, The Netherlands
[15]UMRI Sciences Agronomiques et Procédés de Transformation, Institut National Polytechnique Félix Houphouët-Boigny (INP-HB), Yamoussoukro, Côte d'Ivoire
[16]NOAA Geophysical Fluid Dynamics Laboratory, Princeton, New Jersey, USA
[17]Department of Earth System Science, University of California, Irvine, Irvine, CA, USA.
[18]College of Urban and Environmental Sciences, Peking University, China
[19]The Joint Graduate School of Energy and Environment, King Mongkut's University of Technology Thonburi, Bangkok, Thailand
[20]Barcelona Supercomputing Center, Barcelona, Spain
[21]Universitat Politècnica de Catalunya, Barcelona, Spain
[22]NASA Ames Research Center, Moffett Field, CA, USA
[23]Deutsches Zentrum für Luft- und Raumfahrt, Institut für Physik der Atmosphäre, Oberpfaffenhofen, Germany
[24]European Centre for Medium-Range Weather Forecasts, Bonn, Germany
[25]Netherlands Institute for Space Research (SRON), Leiden, The Netherlands
[26]Department of Environmental Science, Hindu College, University of Delhi, Delhi, India
[27]Department of Environmental Studies, Zakir Husain Delhi College, University of Delhi, New Delhi, India
[28]Hub of Talents on Air Pollution and Climate (HTAPC), Thammasat University, Pathum Thani, Thailand
[29]UK Centre for Ecology & Hydrology, Edinburgh, UK.
[30]School of Chemistry, University of Edinburgh, Edinburgh, UK
[31]WFSRU Kasetsart University, Thailand





[32]Center for Climate Systems Research, Columbia University, New York, NY, USA
[33]NASA Goddard Institute for Space Studies, New York, NY, USA
[34]Cooperative Institute for Research in Environmental Sciences, University of Colorado Boulder
[35]Lancaster Environment Centre, Lancaster University, Lancaster, UK
[36]Center for Policy Research on Energy and Environment, School of Public and International Affairs, Princeton University, Princeton, NJ 08544, USA
[37]Department of Atmospheric Sciences, Rosenstiel School, University of Miami, Miami, FL

*Correspondence to*: Cynthia H. Whaley (Cynthia.whaley@ec.gc.ca)

**Abstract.** Open biomass burning has major impacts globally and regionally on atmospheric composition. Fire emissions include particulate matter, tropospheric ozone precursors, greenhouse gases, as well as persistent organic pollutants, mercury and other metals. Fire frequency, intensity, duration, and location are changing as the climate warms, and modelling these fires and their impacts is becoming more and more critical to inform climate adaptation and mitigation, as well as land management. Indeed, the air pollution from fires can reverse the progress made by emission controls on industry and transportation. At the same time, nearly all aspects of fire modelling – such as emissions, plume injection height, long-range transport, and plume chemistry – are highly uncertain. This paper outlines a multi-model, multi-pollutant, multi-regional study to improve the understanding of the uncertainties and variability in fire atmospheric science, models, and fires' impacts, in addition to providing quantitative estimates of the air pollution and radiative impacts of biomass burning. Coordinated under the auspices of the Task Force on Hemispheric Transport of Air Pollution, the international atmospheric modelling and fire science communities are working towards the common goal of improving global fire modelling and using this multi-model experiment to provide estimates of fire pollution for impact studies. This paper outlines the research needs, opportunities, and options for the fire-focused multi-model experiments and provides guidance for these modelling experiments, outputs, and analysis that are to be pursued over the next 3 to 5 years. It proposes a plan for delivering specific products at key points over this period to meet important milestones relevant to science and policy audiences.

# 1 Introduction

Open biomass burning (BB), which includes wildland fires and agricultural burning (often called "fires" hereafter), has major impacts on global and regional atmospheric chemistry, climate, air quality and the health of ecosystems, via emissions of air pollutants and greenhouse gases, their long-range transport, and their deposition. Fire emissions include particulate matter;, tropospheric ozone precursors, such as nitrogen oxides ($NO_x$), volatile organic compounds (VOCs) and carbon monoxide (CO); long-lived greenhouse gases such as methane, nitrous oxide, and carbon dioxide; persistent organic pollutants; mercury and other metals. While contributions to poor air quality from industrial and transportation sources are decreasing in many parts of the world due to emission controls, fires are a growing contributor to elevated air pollution episodes. Fire frequency, intensity, duration, and location are changing as the climate warms (UN, 2022; Cunningham et al,





2024), and understanding and modelling these changes to fire regimes and their impacts is becoming more and more critical for climate adaptation and mitigation. At the same time, nearly all aspects of fire modelling – such as emissions, plume injection height, long-range transport, and plume chemistry – are highly uncertain. We propose a multi-model, multi-pollutant, multi-regional study to improve the understanding of the uncertainties and variability in fire atmospheric science and its impacts, in addition to providing quantitative estimates of the air pollution and radiative impacts of biomass burning.

The proposed study (herein referred to as HTAP3-Fires) is being planned under the auspices of the Task Force on Hemispheric Transport of Air Pollution (TF HTAP, http://htap.org), an expert group organized under the Convention on Long-Range Transboundary Air Pollution (UN, 1979), to improve understanding of the intercontinental flows of air pollutants, including aerosols and their components, ozone and its precursors, mercury and other heavy metals, and persistent organic pollutants. TF HTAP has an interest in understanding the relative contribution of fires as compared to other sources, to air pollution impacts on health, ecosystems, and climate at the regional to global scale. TF HTAP is also well-positioned to bring together the multi-disciplinary, international modelling and fire science communities to work towards the common goal of improving global modelling of air pollutants released from fires. Although initiated under TF HTAP, this paper and the plan presented herein is intended to reflect the interests of this broader community and to facilitate communication and coordination between a variety of related ongoing activities and new activities that may be initiated as part of this community plan.

This paper outlines the research needs (Section 2), opportunities (Section 3), and options (Section 4) for improving understanding of the climate, air quality, and toxicological impacts of fires and identifies specific research activities and modelling products (Section 5) that could be pursued over the next 3-5 years. Section 5 proposes a plan for delivering specific products at key points over this period to meet important milestones relevant for science or policy audiences.

## 2. Motivation: Science Policy Questions

Several open online meetings were organized by TF HTAP in 2022 and 2023 to identify policy-relevant science questions that could be explored in a study of the transboundary air pollution impacts of fires. The questions identified through those meetings have been subsequently refined into the subsections below. The stated questions are not an exhaustive compilation, but the questions do provide important motivation and direction for the HTAP3-Fires multi-model experiments.

### 2.1 Transboundary transport of fire-emitted compounds

- What are the impacts of fire emissions on air quality, human health, ecosystems, and climate at different scales, from near- to far-fields?



- What is the role of transboundary movements of fire plumes in impacting atmospheric composition in different regions? And how will the absolute and relative magnitudes of these contributions change over time?
- How does the location or seasonality of large fire events within regions affect the long-range transport potential? And how might these locations change over time with land use and climate change?
- How do plume dynamics and near-fire chemical transformations (e.g. sequestration of $NO_x$ in peroxyacetyl nitrate (PAN), formation of secondary organic aerosols) affect the long-range transport potential and downwind impacts?
- Do different fire types (e.g. agricultural waste burning and wildland fires) have different extents of long-range
  transport? What are their relative contributions to regional air pollution?

## 2.2 Fire variability and uncertainty

- What is the range of variability and uncertainty of the results from multiple models' simulations?
- How do model differences in physical and chemical processes manifest in the varied impacts of climate forcing and health that are due to fire emissions?
- Are there certain fire-related parameterizations that perform particularly well against observations?
- What are key model parameters that require improved observational constraints to reduce uncertainty?
- What is the impact of different fire emissions inputs on atmospheric concentrations?
- How sensitive are model results to prescribed fire emissions versus prognostic (interactive fire modules that are coupled to climate) emissions?

## 2.3 Similarities and differences between different pollutants

- What is the contribution of fires to atmospheric concentrations of different air pollutants?
- How do the footprints of different pollutants differ and what are the principal drivers of those differences?
- How much do model source-receptor relationships differ based on initial pollutant focus? (e.g. comparison of different model types)
- How do fire emissions interact chemically with other anthropogenic emissions in the atmosphere?

## 2.4 Questions identified by the research community, but that are beyond the scope of this study

- What are the implications of potential regional changes in prescribed burning, fire suppression policies, and other fire management strategies?
- What is the impact on transboundary smoke from local fire management policies?
- What impact does pyrocumulonimbus have on long-range transport of fire emissions? How often and where does pyrocumulonimbus occur and will they become more frequent with climate change?





- What emissions result when wildfires consume buildings and other infrastructure in the wildland-urban interface? What are the health impacts of built-environment burning?

- How much do fires with small burned areas that are not detected by satellite observations influence the fire emissions amount and composition?

## 3 Scope and background information

The scope and further motivation for this undertaking are defined in this section, partially informing the multi-model experiment design that will appear in Section 5, including the model output table (Section 5.4).

### 3.1 Pollutants of interest

Fires emit all the pollutants that the Convention on Long-Range Transboundary Air Pollution (CLRTAP) is concerned with. This study is an opportunity to address all pollutants with the common emission source of open burning. Below is additional information on these pollutants in the context of fires and this modelling study.

#### 3.1.1 Tropospheric ozone and its precursors

Tropospheric ozone ($O_3$) is both an air pollutant detrimental to human health and vegetation, and a short-lived climate forcer
(SLCF) (Monks et al., 2015). $O_3$ is not emitted directly, but rather formed through photochemical processes involving nitrogen oxides ($NO_x = NO + NO_2$), hydrocarbons, such as volatile organic compounds (VOCs), methane ($CH_4$), and carbon monoxide (CO). This chemistry evolves in fire plumes: freshly emitted plumes, typically containing a lot of particulate matter, may suppress $O_3$ formation due to low-light conditions or heterogeneous chemistry, whereas aged fire plumes may produce $O_3$ more efficiently (e.g., Real et al., 2007). Due to a large quantity of VOC emissions from biomass burning, $O_3$
formation in wildfire plumes is generally NOx-limited. However, when VOC-rich smoke plumes are transported into NOx-rich urban pollution, $O_3$ formation may be enhanced.

The overall impact of fires on $O_3$ concentrations remains highly uncertain. While $NO_x$ is short-lived, it can be transported long distances in the form of PAN (a reservoir for sequestering $NO_x$ and $HO_x$ radicals), leading to additional $O_3$ production in downwind regions for moderate smoke plumes, and production increases with plume age (Jacob, 1999; Lin et al., 2010;
Jaffe and Wigder, 2012; Fiore et al, 2018). At high smoke levels, $O_3$ production can be suppressed, due either to heterogeneous chemistry on smoke particles (e.g., Konovalov et al. 2012) or to diminished photolysis rates (Alvarado et al. 2015). Recent field measurements show that emissions of $NO_x$ and HONO in wildfire plumes are rapidly converted into more oxidized forms such that $O_3$ production in wildfire plumes becomes rapidly $NO_x$-limited (Juncosa Calahorrano et al., 2021; Xu et al., 2021). After a few daylight hours, 86% of the total reactive oxidized nitrogen species ($NO_y$) is in the forms
of PAN (37%), particulate nitrate (27%), and gas-phase nitrates (23%) (Juncosa Calahorrano et al., 2021). When a VOC-rich smoke plume mixes into a $NO_x$-rich urban area, it can also create an environment for enhanced $O_3$ production (Liu et al.,





2016; Gao and Jaffe, 2020). The net impact of fires on regional and extra-regional $O_3$ therefore depends on the emission of a range of precursor species and their chemical transformation in fresh and aged wildfire smoke plumes. Previous HTAP assessments (HTAP1 and HTAP2) have shown that ground-level $O_3$ is significantly influenced by long-range transport at the

hemispheric scale and have demonstrated the utility of a large ensemble of models for quantifying these effects and their uncertainty (Fiore et al., 2009). While fires contribute only a small amount to annual average ground-level $O_3$ in the major northern hemisphere receptor regions, they can be important episodically, and may become more important with global warming and reduction of traditional anthropogenic emissions.

The 1999 CLRTAP Gothenburg Protocol (GP, EMEP, 1999) as amended in 2012 regulates the emissions of $O_3$ precursors in

member states. In a recent review, it was concluded that current air quality legislation in the United Nations Economic Commission for Europe (UNECE) region is not sufficient to meet the long-term clean air objectives of CLRTAP. In support of the CLRTAP response to the recent GP review, TF HTAP is currently organising a new set of multi-model experiments (HTAP3) aimed at quantifying the contribution of long-range transport to ground-level $O_3$ in all world regions from remotely emitted $O_3$ precursors, including from fire emissions (the "Ozone, Particles, and the deposition of Nitrogen and Sulfur", or

HTAP3-OPNS project). To avoid duplication of effort, the model runs contributing to both exercises will be harmonised as much as possible (e.g., using common emission datasets and simulation years).

### 3.1.2 Methane

$CH_4$ is the second most important greenhouse gas after $CO_2$ and modulates the chemistry of many other air pollutants via its impact on atmospheric concentrations of the hydroxy radical (OH). It is also involved in tropospheric $O_3$ photochemistry

(Sec 3.1.1). In addition to $CH_4$ being directly emitted from biomass burning, $NO_x$, CO, and NMVOCs emitted by fires have the potential to alter regional and global OH concentrations, thus influencing the atmospheric lifetime of $CH_4$ (e.g., Naus et al., 2022). Modelling studies suggest significant suppression of global OH concentration following enhanced CO emissions from extensive wildfires in Southeast Asia during El Niño events (Duncan et al., 2003; Manning et al., 2005; Rowlinson et al., 2019). Butler et al. (2005), and Bousquet et al. (2006) both found that this change in global OH significantly contributed

to the observed increase in global $CH_4$ concentration during the 1997 El Niño fires. The influence of fires on global OH appears to depend on the location of the fires. Leung et al. (2007) showed that the CO emissions from extensive boreal fires in 1998 did not significantly lower global OH, and thus did not significantly contribute to enhanced $CH_4$ growth. Rowlinson et al. (2019) showed that the increase in $CH_4$ lifetime induced by El Niño-related fires in the tropics offsets an El Niño-driven reduction in $CH_4$ lifetime caused by changes in humidity and in atmospheric transport.

Extreme fires and fire seasons may lead to increased $CH_4$ emissions from wildland fires. For example, the 2020 extreme fire year in California accounted for approximately 14% of the state's total $CH_4$ budget, including all anthropogenic $CH_4$ sources (Frausto-Vicencio et al., 2023). Fires in Arctic tundra will also lead to more $CH_4$ emissions in the future, as recent observations in Alaska revealed that previously burned tundra (within 50 years) emit more $CH_4$ than the surrounding landscapes (Yoseph et al., 2023).





### 3.1.3 Particulate Matter


Particulate matter (PM) is emitted in great quantities from fires and is usually the main cause of air quality exceedances during fire episodes. In addition, it has consequences for cloud interactions and radiative forcing. It is comprised of a range of species including black carbon (BC, also known as elemental carbon or soot), primary organic carbon (OC, related to organic aerosol, OA), sulfate ($SO_4$), nitrate ($NO_3$), ammonium ($NH_4$), and crustal material (CM, or dust). Particulate matter

may be emitted directly or can be formed as secondary aerosols through gas-to-particle conversion. Secondary organic aerosol (SOA) is particularly important in the context of long-range transport (see Section 3.2.5). If smoke is transported through a cloudy boundary layer, aqueous-phase processing can also facilitate the transformation of $SO_2$ gas into sulfate, with consequences for cloud interactions (e.g., Dobracki et al, 2024). The chemical and radiative properties, as well as cloud interactions are all dependent on the chemical composition, size, and vertical distributions of the particulate matter (e.g.,

Huang et al., 2012). BC accounts for about 10% of smoke plume mass and is the most critical contributor to aerosol radiative forcing (RF) (Veira et al., 2016). In contrast to other aerosol components, BC introduces a radiative warming into the Earth's climate system (Section 3.2.2). Compared with BC from fossil fuel combustion, BC from biomass burning has generally larger particle sizes, more and thicker coated particles, and more absorption per unit mass (Schwarz et al., 2008).

### 3.1.4 Mercury

Mercury (Hg) is a potent neurotoxin that bioaccumulates in the environment, endangering human health, wildlife, and ecosystems. Wildfires release mercury from plants and soils into the atmosphere, where it may be carried and deposited over great distances, contaminating water bodies and terrestrial ecosystems (Obrist et al., 2018; Chen and Evers, 2023). The Minamata Convention on Mercury (UN, 2013), a worldwide convention enacted in 2013, seeks to safeguard human health and the environment against mercury's negative effects. It examines the complete life cycle of mercury, including extraction,

trading, use, and emissions, emphasizing the need of reducing mercury pollution internationally. A third set of multi-model experiments being organized under HTAP3, known as the Multi-Compartmental Mercury Modelling and Analysis Project (HTAP3-MCHgMAP), is aimed at attributing trends in environmental mercury concentrations to changes in primary mercury emissions and releases or to changes in other drivers or processes (Dastoor et al, 2024). All three sets of HTAP3 experiments (Fires, OPNS, and MCHgMAP) will aim to harmonise inputs and experimental designs as much as possible and

avoid duplication of effort.

### 3.1.5 Persistent organic pollutants

Persistent organic pollutants (POPs), which are synthetic chemicals that are also bioaccumulative, toxic and subject to long-range transport. POPs that have been trapped through wet and dry deposition by trees and shrubs (Su and Wania 2005, Daly et al. 2007) can be re-released during a wildland fire. The high temperature and vertical winds of wildland fires can

remobilize POPs from fuels such as leaves and needles and the forest soil, which otherwise act as a sink for POPs. Eckhardt





et al. (2007) reported record high concentrations of polychlorinated biphenyls (PCBs) at the Arctic station of Zeppelin (Svalbard) in a forest fire plume after a transport time of 3-4 weeks. Many atmospheric models do not simulate POPs, however, several POPs models exist, with some listed in Table A.2.

The UNEP Stockholm Convention on POPs provides the framework for global regulation and monitoring of POPs since 235 2004. However, many POPs, e.g. polychlorinated biphenyls, dichlorodiphenyltrichloroethane and its degradation products (DDTs), other organochlorine pesticides, polybrominated diphenyl ethers (PBDEs), and per- and polyfluoroalklyl substances (PFASs), have been in use for decades before they were regulated. While most legacy POPs in air are declining globally (Wong et al. 2021, Shunthirasingham et al. 2018, Kalina et al. 2019), increasing trends are observed for chemicals of emerging concern, e.g. PFASs (Wong et al. 2018, Saini et al. 2023).

Dioxins are one class of POPs that are formed during incomplete combustion processes. Dioxins are emitted from waste incineration, industrial and residential combustion of fossil fuels, and biomass burning. Global gridded emission inventories are now available for dioxins (EDGAR at http://edgar.jrc.ec.europa.eu; and Song et al., 2022). Compared to the early 2000s, global dioxin emission reduced by 26% in the late 2010s, attributable to emission mitigations in upper- and lower-middle income countries. However, the declining trend of dioxin emissions over the past decades terminated from the early 2010s 245 due to increasing significance of wildfire induced emissions in the total emission. The highest levels of dioxin emissions ( expressed as polychlorinated dibenzodioxins/dibenzofurans (PCDD/Fs)) were identified in East and South Asia, Southeast Asia, and part of Sub-Saharan Africa. In East and South Asia, growing dioxin emissions are attributed to industrialization, whereas wildfire is a major contributor to high dioxin emissions in Southeast Asia and Sub-Saharan Africa.

### 3.1.6 Polycyclic aromatic hydrocarbons

Polycyclic aromatic hydrocarbons (PAHs) are organic pollutants primarily generated by incomplete combustion. PAHs are of concern because their concentrations have remained stable despite global emission reductions. PAHs exist in both gas and particulate phase in the atmosphere, allowing them to undergo long-range transport to remote locations (Muir and Galarneau, 2021; Zhou et al., 2012). PAHs are regulated under the UNECE Aarhus Protocol on POPs in the CLRTAP (Yu et al, 2019), yet are still observed in pristine, remote areas, such as the Arctic and Antarctic regions. The long-range atmospheric 255 transport of PAHs has been extensively investigated and partly attributed to sources in global emission inventories (e.g., PEK-FUEL at http://inventory.pku.edu.cn/ and EDGAR at http://edgar.jrc.ec.europa.eu). Further efforts to update global monthly PAH emissions from wildland fire sources from 2001 to 2020 use carbon stock data up to 2020 based on satellite remote sensing (Luo et al., 2020; Song et al. 2022). The new inventories improve modelling of wildfire-induced PAH levels and trends particularly in the Arctic, Sub-Saharan Africa, Southeast Asia, and South America. In the Arctic, source-tagging 260 methods have identified local wildfire emissions as the largest sources of benzo[a]pyrene (BaP), a congener of PAHs with high carcinogenicity, accounting for 65.7% of its concentration in the Arctic, followed by wildfire emissions of Northern Asia. Wildland fires account for 94.2% and 50.8% of BaP levels in the Asian Artic during boreal summer and autumn, respectively, and 74.2% and 14.5% in the North American Arctic for the same seasons (Song et al., 2022).





### 3.1.7 Other metals and trace elements

Biomass-burning aerosols also contains a large variety of metals and other trace elements (Perron et al. 2022). The source can be the vegetation consumed and/or surrounding soils entrained into plumes by strong pyroconvective updrafts (Wagner et al., 2018; Hamilton et al., 2022), or mixing of BB aerosol emissions into advecting dust plumes, as happens in sub-Saharan Africa (Quinn et al., 2022). Entrained soil dust is estimated to be the major (two-thirds) source component for the iron contained in smoke plumes (Hamilton et al., 2022), with other elements needing further investigation. Many of these

elements are important components for biogeochemical cycles, human health impacts, and/or aerosol RF.

The mass of iron emitted by fires is particularly important to quantify because iron is a limiting nutrient in many open ocean regions, playing an important role in $CO_2$ sequestration, particularly in the southern oceans through increasing phytoplankton primary productivity (Tang et al., 2021; Hamilton et al., 2020).

Other nutrients (e.g., phosphorus) are also emitted from fires in sufficient quantities to warrant deeper understanding of their

fluxes and related impact assessment on terrestrial and marine biogeochemical cycles. For example, African fires have been identified as an equal source to African dust in terms of the intercontinental transport of phosphorus to the Amazon Rainforest (Barkley et al. 2019). There is also growing evidence that increasing United States (US) fire activity is impacting downwind freshwater ecosystems through depositing phosphorous (Olsen et al., 2023).

One practical issue in determining the impact of changes in fire activity on metal aerosol emission and deposition fluxes is

quantifying the contribution of fire to the atmospheric loading of a given metal. There are many other sources of metals to the atmosphere, including mineral and anthropogenic dust, fossil fuels and vehicular transport, metal smelting and mining, and volcanoes to name a few (Mahowald et al., 2018; Hamilton et al., 2022). Once sources become well-mixed in the atmosphere it becomes much more difficult to trace their individual source contributions. One potential avenue in "fingerprinting" the fire source contribution is the use of metal isotopes. In general, different metal sources have different

isotopic fractionations (Fitzsimmons and Conway, 2023) and this difference in aerosol characteristic has been used successfully to differentiate iron aerosol between dust and anthropogenic sources (Conway et al., 2019). However, there is currently no data on the iron isotopic signature of fire, so that aspect is beyond the scope of this study.

### 3.2 Impacts from fires

### 3.2.1 Human health

Densely populated areas like Southeast Asia, North America, and the Mediterranean experience episodes of intense air pollution from wildfires exceeding the ambient air quality standards that last multiple days or weeks on a regular basis (Liu et al., 2015; Jaffe et al., 2020; Dupuy et al., 2020; and see Supplement, Sections S1 for further regional discussions and S2 for acute exposure health impacts). An estimated 339,000 premature deaths per year (interquartile range: 260,000 - 600,000) can be attributed to exposure to wildfire smoke worldwide (Johnston et al., 2012). Xu et al (2023) estimated 9.9 days of

smoke exposure from 2010-2019, a 2.1% increase compared to the previous decade. The impacts are projected to increase





under future climate change (Xie et al., 2022). In many regions of the world, farmers commonly burn crop residues to clear land for crop cultivation. However, these agricultural fires have health implications as air pollution increases (Jones and Berrens, 2021). During peak fire periods, these agricultural fires can contribute more than half of the particulate matter (PM) pollution, even in urban settings (Cusworth et al, 2018; Liu et al., 2018).

Health risk assessment models and air quality health indices are often based on surface level concentrations of $PM_{2.5}$, CO, $O_3$, and $NO_x$. Emissions of $PM_{2.5}$ from fires are of particular health concern, with no known safe $PM_{2.5}$ concentration in air, as noted by the World Health Organization (WHO 2006). Fine particles impact lung functions, encouraging respiratory and cardiovascular mortality and morbidity including asthma and emphysema (Davidson et al., 2005; Lampe et al., 2008; Jain et al., 2014; Reid et al., 2016; Cascio, 2018; Ghosh et al., 2019; Chen et al, 2021; Aguilera et al., 2021; Sonwani et al. 2022;

Gao et al., 2023; Bauer et al., 2024). There is also evidence that wildfire smoke affects mental health (Eisenman et al., 2022; To et al., 2021), such as due to displacement and smoke exposure following wildfires which can lead to increased cases of anxiety and post-traumatic stress disorder (e.g., Humphreys et al., 2022).

An additional consideration is how smoke influences the structure of the boundary layer and thus the concentration of pollutants that people are exposed to. Fire aerosols, by cooling the surface and reducing boundary layer turbulence (Section

3.2.2), can suppress mixing of air in the boundary layer, effectively increasing pollution exposure at the surface (Bernstein et al., 2021). This effect has been studied extensively in polluted urban environments, but its importance for fires, where the composition of aerosol may be substantially different, remains unclear.

Finally, the chemical composition of the PM influences its health impacts. For example, benzo(a)pyrene, the most toxic congener of 16 parent PAHs, has been linked to high lifetime cancer risk from inhalation. Knowledge of PM size distribution

(e.g. Sparks and Wagner, 2021), and chemical composition is essential for understanding health impacts, thus, motivating our multi-pollutant approach to these model simulations.

### 3.2.2 Climate and radiative forcing (RF)

While wildland fires have long been considered a natural and relatively carbon-neutral component in the Earth system ($CO_2$ emitted during burning is reabsorbed as the forest regrows), land use change and anthropogenic climate change have caused

the frequency and intensity of fires to rapidly change, potentially altering the global carbon budget. The magnitude of preindustrial fires 'is highly uncertain, propagating into large uncertainties in aerosol RF estimates and thus understating how climate has evolved over the Industrial Era is highly uncertain by a factor of 4 (Hamilton et al, 2018; Wan et al, 2021; Mahowald et al. 2023).

Fire emissions have diverse effects on the climate. In addition to the direct effects from released greenhouse gases and

aerosols, additional indirect effects arise from the formation of tropospheric $O_3$, reduction in lifetime of $CH_4$ by enhancing tropospheric oxidation capacity, and changes in stratospheric water vapor caused by responses of the atmospheric chemistry. Co-emitted $SO_2$ can also become converted to $SO_4^{2-}$, an effective cloud nucleator, thereby affecting cloud lifetime (Dobracki et al., 2024). The aerosols have indirect (microphysical) and semi-direct (radiative) impacts on cloud fields and large-scale





circulation (Ding et al., 2022; Diamond et al., 2020; Adebiyi and Zuidema, 2018). Short-term radiative effects of smoke on

surface wind, temperature, moisture, and precipitation can also substantially enhance fire emissions and weaken smoke dispersion (Grell et al, 2011; Huang et al., 2024). Snow and ice albedos also change dramatically when fire-emitted black and brown carbon are deposited. Additionally, indirect effects on biogeochemistry result from wildfire emissions (Sections 3.1.6 and 3.2.3).

The RF from fire plume components are summarized in Table 1. Though, most studies focus on specific components or

regional RF of wildfire emissions only (e.g., Mao et al., 2012; Chang et al., 2021, Mubarak et al., 2023). However, Ward et al. (2012) conducted a comprehensive global analysis of wildfire emission's RF, encompassing all components.

**Table 1:** Summary of present-day RF from specific fire plume components.

| Fire emission component | RF (W/m2) | Comments |
|---|---|---|
| Tropospheric $O_3$ | 0.03 to 0.05 | Dahlman et al., 2011, Ward et al, 2012. Depends heavily on the emissions from other sources, emission location and plume height (e.g. Naik et al., 2007; Paugam et al., 2016). |
| Aerosol direct effect | -0.20 to 0.25 | Rap et al., 2015; Tian et al, 2022. Depends on uncertainties in BC absorption, and height of the smoke plume. |
| Aerosol indirect effect | -1.11 to -0.09 | Tian et al, 2022; Rap et al., 2015. Depends on background conditions. |

The large range in aerosol indirect effect heavily depends on the background conditions. Aged smoke is an excellent cloud condensation nuclei (e.g., Kacarab et al, 2019), but increasing cloud condensation nuclei from other emission sources can reduce the RF from wildfire emissions (e.g. Ward et al., 2012; Hamilton et al., 2019), which is a general feature for natural emissions (Spracklen et al, 2013). A reduction of anthropogenic emissions in the future could increase the effects of natural emissions (see example for tropospheric ozone by Mertens et al., 2021). The estimate of the aerosol albedo effect also varies

in sign, but the magnitude is in general rather small compared to the indirect aerosol effect (Tian et al., 2022). The height of the fire plume influences its RF, and recent studies suggest a large climate impact of fire emissions that rise into the stratosphere (Stocker et al., 2021, Damany-Pearce et al., 2022). Moreover, new measurement data indicate a larger warming potential of the aerosol emissions from fires, in part because of low single-scattering albedos resulting from a high fraction of black-carbon containing particles, and relatively low OA:rBC mass ratios (Dobracki et al., 2023) might help to reduce

these discrepancies between the models (Zhong et al., 2023).

### 3.2.3 Ecosystems

Fires impact land cover, runoff/infiltration, soil erosion, and water quality, via reducing water use by plants and increasing soil hydrophobicity. The impact depends on the surface (topography, vegetation type, soil type) and fire properties as well as the quantity and intensity of precipitation following the fires. For example, high forest fire counts in India have been shown





to decrease the soil moisture content, evapotranspiration, and normalized difference vegetation index (Jain et al, 2021). Further regional discussions can be found in the Supplement, in Section S1. Note that for some regions, actions have been taken to reduce such impacts which may or may not be accounted for or represented well in models.

Fires can also positively or negatively impact aquatic and land ecosystems nearby and afar via deposition. Specifically, fires can impact downwind marine ecosystems if nutrient deposition is sufficient to alleviate nutrient limitation in the surrounding

waters (Hamilton et al. 2022). For example, Siberian fires were recently linked to anomalously high phytoplankton growth in the Arctic Ocean through the additional atmospheric supply of nitrogen (Ardyna et al. 2022). Ozone produced from fire and other emissions can reduce the productivity of $O_3$ sensitive ecosystems, perturbing biogenic emissions.

The estimated deposition fluxes depend highly on the models' deposition schemes and vary by chemical species and surface types (e.g., Tan et al., 2018; Huang et al., 2022). Through radiative impacts which are only accounted for in some models,

fires can perturb numerous variables relevant to the calculation of deposition velocity/coefficient and secondary pollutant formation (e.g., Huang et al. 2024, for a Canadian wildfire event in 2023 that enhanced $O_3$ and nitrogen deposition in the eastern US).

### 3.2.4 Socioeconomics and fire management decisions

In cases of forest fires that encroach on the wildland-urban interface, people are forced to evacuate or permanently relocate

their homes. High fatalities of residents (e.g., Molina-Terrén et al. 2019), firefighters, and fauna; severe air pollution ranging over a few to thousands of kilometers; and huge economic losses from property damages, national park closures, tourism and recreational activity curbs, highway blocks, air travel diversions, and forest-based livelihood losses (e.g., Psaropoulos, 2021) result from large scale, recurrent forest fires (Bowman et al. 2011).

Catastrophic wildfires around the world are increasingly more frequent and hazardous. For example, in the United States,

fire-loss events increased from an average of 1.5 events per decade from 1980-1999 to 7 per decade from 2000-2019, costing the nation a cumulative USD \$10 billion and USD \$75 billion, respectively (Smith et al. 2020). Few studies have reported on the increasing socioeconomic impacts and diversity of people and communities being affected (Moritz et al. 2014; Bowman et al. 2017). Further studies denoting the dollar-cost of fire events include Masters (2021) for the 2019-2020 Australia fires and Wang et al. (2021) for the 2018 California fires. Additional regional discussions can be found in the Supplement,

Section S1.

The Wildland Urban Interface (WUI) is the area where human development meets or intermixes with wildlands (Stewart et al., 2007; Platt, 2010). The WUI area plays a critical role in wildfire management because increased human availability in the WUI leads to more human caused ignitions, wildfires in this area pose a greater risk to structures and lives, and WUI fires are harder to manage yet must be suppressed (Choi-Schagrin, 2021). The demographics of the WUI are regionally-

dependent (e.g., Wigtil et al., 2016; Davies et al., 2018; Tang et al., 2024), and are changing with time, as housing costs (Greenberg, 2021), and immigration (Shaw et al. 2020) evolve over time.





Moreover, some studies focused on environmental justice describe various impacts to, and the social vulnerability of, different communities. Wildfires preferentially impact US regions with lower populations of minorities and higher populations of elderly (Masri et al., 2021). Elderly populations are particularly vulnerable to the effects of fire (Masri et al.

2021; Liu et al. 2015; Murphy and Allard, 2015). Indigenous communities also have high vulnerability, because they are disproportionately located in areas of high fire risk (Davies et al. 2018).

Land management decisions have an important role in determining ecological and socio-economic pathways. Prescribed or controlled burning is an important tool within holistic land management plans for enhancing ecosystem resilience, biodiversity conservation, plant response, air quality, and carbon sequestration. Each of these benefits are expanded upon in

the Supplement, Section S3, with the general conclusion that collaboration with local communities, incorporation of traditional ecological knowledge, and adaptive management techniques guarantee that land management decisions are consistent with sustainable practices. Further research beyond the scope of this study is needed to incorporate these kinds of land management decisions into fire emissions scenario inputs for atmospheric models.

### 3.2.5 The role of atmospheric long-range transport

Long-range transport of fire-related pollutants makes open biomass burning relevant for regions that are not typically impacted by widespread, frequent or intense fires. For example, recent Canadian 2018 and 2023 fires were reported to cause high PM and $O_3$ pollution episodes in the US (e.g., Xie et al., 2020; Lin et al., 2024; Huang et al., 2024), and these plumes can reach Europe through long-range cross-Atlantic transport (Real et al., 2007; Alvarado et al., 2020; CAMS, 2023). In tropical regions, prevailing easterlies and the African Easterly Jet South (Adeyemi and Zuidema, 2016) can readily transport

biomass-burning aerosol from Africa to South America (Holanda et al. 2020). The biomass-burning aerosol interactions with a large subtropical low cloud deck vary microphysically and radiatively with the vertical colocation of aerosol and cloud (Kacarab et al., 2020; Zhang and Zuidema, 2019; 2021). Smoke is also an annual occurrence in northern Thailand and upper Southeast Asia, transported regularly to southern China and Taiwan. At even larger scales, global teleconnections such as the El Niño-Southern Oscillation allowing variability in precipitation, and thus, Indonesian peat fires to impact emission

loadings as far away as equatorial Africa (Doherty et al., 2006; Lin et al., 2014). Smoke also impacts the southeast Asian monsoon through increasing the low cloud coverage (Ding et al., 2021).

Long-range transport depends on many factors including, but not limited to source proximity, plume height, synoptic weather conditions, atmospheric chemistry and deposition rates. Long-lived primary pollutants such as CO may be transported on a hemispheric scale, while short-lived species such as PM and $NO_x$ typically affect a much smaller region.

However, the formation of secondary pollutants within the plume introduces substantial uncertainty into the broader atmospheric impacts of fires. In particular, the formation of longer-lived pollutants such as $O_3$, PAN and secondary fine particles can substantially impact atmospheric composition over intercontinental distances, documented in both observational and modelling studies (Real et al., 2007; Lin et al., 2024). The timing and magnitude of secondary pollutant formation in transported plumes strongly influences the health and ecosystem impacts of distant downwind regions and





introduce much uncertainty in our assessment of these impacts. Sensitivity experiments with atmospheric chemistry transport models, constrained with estimates of formaldehyde, a by-product of secondary organic aerosol that can be detected from space (Zhong et al., 2023; Alvarado et al., 2020) are important for understanding the long-range impact of fire-related primary and secondary pollutants on receptor regions.

As nations implement more stringent air quality targets, long-range transport will start to play an increasingly important role
in determining if these targets are met. The multi-model study proposed here will include regional emissions perturbation experiments (Sections 4.5 and 5.3) to quantify the long-range impacts on local atmospheric composition.

### 3.3 Leveraging recent and ongoing efforts

Several distinct scientific communities are addressing fire research and applications in line with their specific objectives. Table A.1 of the Appendix lists the recent and ongoing efforts in the community that are complementary, but not duplicating
the research outlined in this paper. For example, the IGAC BBURNED activity hosted a workshop in November 2023 to assess current global biomass burning emissions datasets and recommend one as the baseline fire emissions dataset for this work (Sections 5.2 and 5.3). The Arctic Monitoring and Assessment Programme (AMAP) SLCF expert group may utilise the model output from this work for a future Arctic-focused biomass burning report. A further example is the Climate Model Intercomparison Projects, CMIP6 and CMIP7 activities: CMIP6 and FireMIP included simulations from dynamic vegetation
models with interactive fire modules provided future fire emissions for different climate scenarios as input for this work (Section 4.2.3 and 5.2), and AerChemMIP2 being planned for CMIP7 will include fire-focused simulations for their aerosols and gas chemistry climate impacts.

## 4. Discussion of modelling options

In this section, we establish the range of model types expected to participate, and then discuss different options for model
inputs, such as emissions and driving meteorology. We also discuss what kinds of simulations could be carried out to answer the science policy questions of Section 2. Final guiding decisions on all of these topics are provided in Section 5.

### 4.1 Model types and scope

Models suitable for exploring the local, regional and global impacts of fires have a wide range of different geographic and temporal scales and resolutions. Models of atmospheric processes have widely differing treatments of chemical complexity
and differ in their vertical and horizontal extent. Some models incorporate physical processes to simulate their own meteorology, which may be nudged to match meteorological reanalyses, while others are driven directly with reanalysis data. More complex models may incorporate other Earth system components including the land surface and vegetation (which may or may not be interactive), ocean exchange (and sometimes biogeochemistry), and the cryosphere. In some models, fire ignition, spread and pollutant emission are explicitly represented, governed by vegetative fuel loading and





meteorology, while in others they are a specified input. This diversity in model types and scope presents a technical challenge in comparing the simulated impacts of fires between models (e.g., Shinozuka et al., 2020; Doherty et al., 2022), but the different approaches and levels of complexity present a valuable opportunity to provide fresh insight into our understanding of fire processes and how they are best represented for specific goals. The models participating may fall into these categories:

• Earth System Models (ESM) or Coupled Chemistry-Climate Models (CCM)

    • Regional or global Chemical Transport Models (CTM)

    • multi-media POPs models

    • Lagrangian Transport Models

    • Reduced-form, surrogate models (e.g., emulators)

• Inverse models (see Section S4 for more information)

The modelling centres in Table A.2 have indicated interest in participating in this study. The characteristics of the models in Table A.2 are taken into consideration for the experimental design.

## 4.2 Available emissions inputs for historical and future simulations

Almost all atmospheric models will require some information about anthropogenic and natural emissions as inputs. In this
section, we discuss available data sets for both historical and future anthropogenic and natural emissions relevant for a global multi-model study. Extricating truly natural from anthropogenic biomass burning is a tricky endeavour that is beyond the scope of this study. For example, while residential wood combustion is considered uncontroversially as anthropogenic, would accidental human ignition of a wildfire be considered natural or anthropogenic biomass burning? Similarly, would wildland fires that are more frequent and intense due to anthropogenic climate change be considered natural or
anthropogenic? For the model design and interpretation of results, we simplify the total fire emissions into those with and without agricultural burning, and classify traditional fossil-fuel emissions as anthropogenic. Agricultural burning appears in both kinds of emissions datasets, so guidance is provided in Section 5.2 on which to use to not double-count those emissions.

### 4.2.1 Historical and future anthropogenic emissions

The HTAP v3 global anthropogenic emissions mosaic (Crippa et al., 2023) covers the time period 2000-2018 at 0.1 x 0.1
degree resolution and monthly temporal resolution. This mosaic inventory is based on the EDGAR 6.1 global inventory and incorporates detailed emissions (for 16 sectors) for $SO_2$, $NO_x$, CO, NMVOC, $NH_3$, $PM_{10}$, $PM_{2.5}$, BC, OC, and four POPs species from several national and regional inventories using the original spatial distributions wherever possible. The REAS v3.2.1 regional inventory is used for Asia (South Asia, East Asia, and South East Asia), the CAMS-REG v5.1 regional inventory is used for Europe, the CAPS S-KU national inventory is used for South Korea, and the official national
inventories of Japan, Canada, and the United States of America are used for the respective geographical zones. Wherever the





respective regional or national inventories did not include specific emission sectors, or wherever these sectors did not include the full set of species provided by EDGAR 6.1, these emissions were gap-filled using EDGAR 6.1. In cases where regional or national inventories included minor sources not present in EDGAR 6.1 (eg., CO, NOx, and SO2 from the solvents sector), these emissions were included in the HTAPv3 mosaic. This inventory is thus a complete and model-ready dataset representing the best available emissions for global and regional model simulations aimed at informing air quality policy. By September 2024, HTAP v3.1 global anthropogenic emissions is expected to be released, which are as above, except covering the years 2000-2020, based on EDGAR v8, and including updated emissions from the regional inventories.

Future scenarios of anthropogenic air pollutant and $CH_4$ emissions are available from the IIASA GAINS integrated assessment model for the period 2015-2050. The scenarios are based on those originally produced in 2021 by IIASA to support the review of the amended Gothenburg Protocol carried out under the Convention on Long-Range Transboundary Air Pollution. The next version of these scenarios, called GAINS LRTAP, will be available from July 2024 and will be used to support HTAP activities aimed at modelling future air quality to inform the CLRTAP policy response to the Gothenburg Protocol review. Three scenarios are provided: CLE (Current Legislation) is based on realistic implementation of existing air quality plans; MTFR (Maximum Technically Feasible Reduction) is based on the same underlying activity data as CLE, but with full implementation of all proven technical measures to abate $CH_4$ and air pollutant emissions regardless of cost effectiveness; and LOW, which builds on MTFR, adding additional structural measures representing climate policies consistent with Paris Agreement goals and dietary changes aimed at reducing emissions from the agriculture sector.

The HTAPv3 historical emissions and LRTAP future scenarios will be used in other concurrent HTAP3 projects (MCHgMAP and OPNS). Use of these emissions datasets would provide consistency across the HTAP3 experiments and would maximize policy relevance of the experiment results. While the historical emissions from HTAPv3.1 and the future scenarios from LRTAP do overlap in time (2015-2020), they have not been harmonised with each other, so do not provide a seamless timeseries of emissions from 2000 to 2050. At present, historical simulations and future simulations should be planned separately. This discontinuity around the present day is consistent with fire emissions themselves (satellite products for the historical period and land model products for the future) and likely also the choice of models (CTMs and specified dynamics for the historical period and free running atmospheric models for the future).

Each modelling centre will need to pre-process the selected emissions datasets to account for vertical profiles and diurnal variations of these emissions. As these processes may differ across models and it may not be possible to harmonize these characteristics, these processes will introduce a source of variability in emissions inputs across models. However, if models use their own default assumptions for vertical and temporal allocation, their methods and assumptions may be reported with their output and taken into consideration in the analysis of outputs.

### 4.2.2 Historical biomass burning emissions

The latest available major global fire emissions datasets are: GFEDv4s, GFASv1.2, FINNv2.5, FLAMBE, QFEDv2.5, GBBEXPv4, and IS4FIRES. Developers of each of these datasets attended and presented their methods at the Fire Emissions



Workshop (FEW2023 at https://www2.acom.ucar.edu/bburned/fire-emission-workshop-virtual-2023; co-hosted by

BBURNED and TF HTAP) in November 2023. Intercomparison studies such as Griffin et al (2023), Pan et al (2020), Wiedinmyer et al (2023), and Liu et al (2020) were also presented there, and the workshop attendees discussed options for which dataset to recommend for consistent baseline fire emissions. An intercomparison tool called FIRECAM (https://globalfires.earthengine.app/view/firecam) was useful for intercomparison. The different methodologies used to estimate fire emissions (e.g. Table 2) account for how and why the emissions results are so different from one another

(Figure 1). The intercomparison studies demonstrated that no one fire emissions dataset performed best for all locations and all pollutants.

**Table 2:** Summary of characteristics of major global fire emissions, adapted from Liu et al (2020). Dash indicates missing value.

|  | Bottom-up | | Top-down | | |
|---|---|---|---|---|---|
| **Fire Emissions dataset:** | **GFEDv4s** | **FINNv1.5, v2.5** | **GFASv1.2** | **QFEDv2.5r1** | **FEERv1.0-G1.2** |
| Horizontal resolution | 0.25º | 1km | 0.1º | 0.1º | 0.1º |
| Near-real time availability | - | ✓ | ✓ | ✓ | ✓ |
| Input satellite fire product | BA + active fire geolocations | Active fire geolocations | FRP | FRP | FRP |
| Peatlands | ✓ | - | ✓ | - | - |
| Cloud-gap adjustment | - | - | ✓ | ✓ | ✓ |
| References | Van der Werf et al. (2017) | Wiedinmyer et al. (2011, 2023) | Kaiser et al. (2012) | Darmenov and da Silva (2013) | Ichoku and Ellison (2014) |






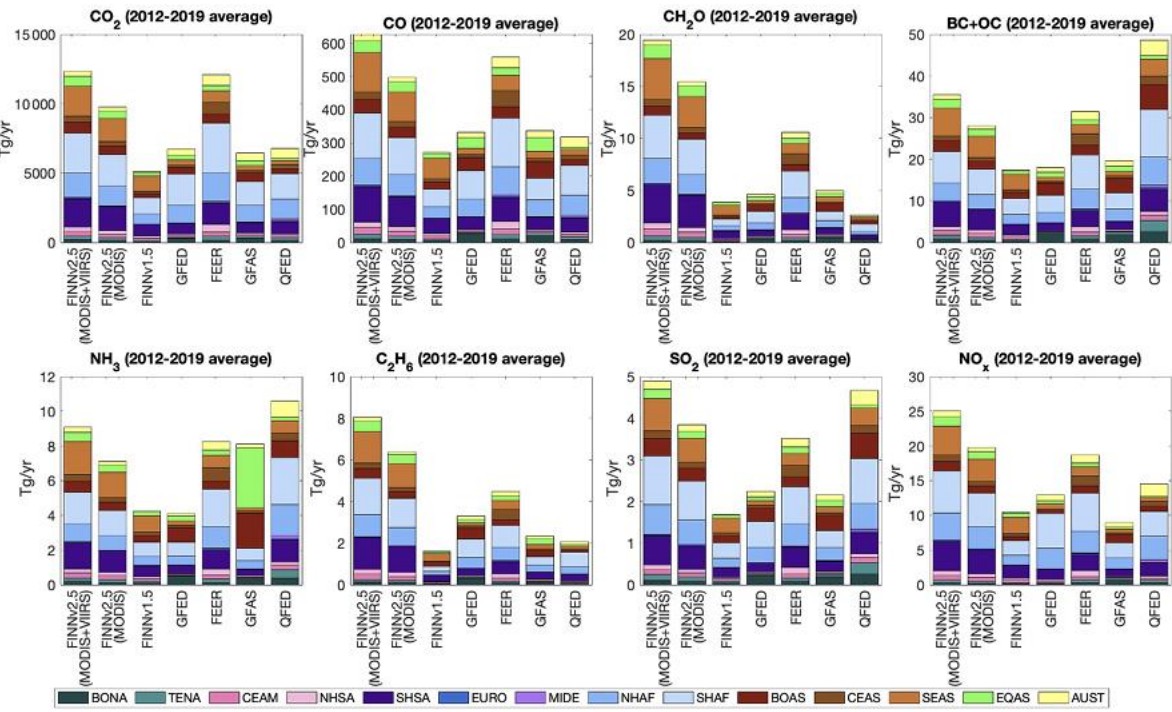

**Figure 1:** Multi-species, multi-regional intercomparison of fire emissions datasets, from Wiedinmyer et al (2023), where regional acronyms and colours are defined in Figure 3c.


*Fire emissions from peat*

Satellite data assimilation studies have shown that emission inventory underestimations may often be due to lack of peat fires. For example, Nechita-Banda et al. (2018) found that incorporating satellite measurements of CO increased CO emissions (compared to GFAS and GFED) from peat fire in Indonesia during the 2015 El Niño event. The ability to account

for emissions from peat fires is a key issue in several regions. Recent work to improve peat fires for Indonesia was done in Kiely et al. (2019). Out of several global fire emissions datasets, only GFASv1.2 and GFEDv4 have tropical peat fires, and only GFASv1.2 contains high-latitude Siberian peatland fires. GFEDv5 emissions will have high-latitude peat fires, but as of this writing, it has not yet been released, nor evaluated. Similarly, a newer version of GFAS (v1.4) is not published or documented yet, though it could have improvements to long-term trends in fire emissions.

Regardless of their inclusion, peat fire emissions are highly uncertain (McCarty et al., 2021). There are different EFs for high-latitude and low-latitude peat fires, given the different vegetation that grows on top, and global maps of peatland are out of date (McCarty et al., 2021). It is also very difficult to detect smouldering (low intensity) peat fires from satellite measurements. That said, the consensus recommendation from FEW 2023 was to use GFASv1.2 based on its inclusion of high-latitude peat fires, ease of adjusting EFs, possibly somewhat better sensitivity in FRP than burnt area, and availability

of information on the diurnal cycle: tropical and mid-latitude peat fires generally have a flat diurnal cycle, apparent in the



FRP observations during daytime and night-time. (e.g. Figure 10 in Kaiser et al. 2012). Diurnal information is directly available in the separately assimilated daytime and night-time FRP in GFASv1.4, but this database has not yet been used to adapt the emission factors.

*Magnitude of Emissions*

Substantial uncertainty arises from estimates of the magnitude and location of emissions. This can be explored through short case study simulations investigating the use of alternative emission datasets, along with comparison of these with observations and baseline model studies. Such sensitivity studies implicitly include differences in resolution and species fractionation (and possibly injection height and timing), as well as fire magnitude and location but nevertheless can provide a useful estimate of uncertainty to fire emissions across the models (Shinozuka et al., 2020; Doherty et al., 2022).

*Timing of Emissions*

Most long-term model studies, such as those performed for CMIP intercomparisons, apply monthly-mean fire emissions rather than considering more temporally resolved emissions that capture the largely episodic nature of fires. The implications of this, either for comparison with surface observations or for regional and global budgets, remain unclear. In addition, there are substantial diurnal cycles in fire intensity, local meteorology and boundary layer dynamics that suggest that the impacts 560 of fires are likely sensitive to the timing of emissions through the day. Observational evidence indicates emerging overnight fires due to increasing drought conditions that challenge the traditional diurnal cycle characterized by 'active day, quiet night' (Luo et al., 2024). These uncertainties can be explored through short studies (one year/several years) that consider (1) monthly mean fire emissions, based on the same set of emissions used at higher temporal resolution in the baseline run, and (2) emissions provided without a diurnal variation in magnitude or injection height.

*Fire emissions of other species: Hg, POPs, PAHs*

Mercury fire emissions are not included by default in most global fire emissions datasets, like GFASv1.2. For the HTAP3 MCHgMAP project, Hg fire emissions are based on FINNv2.5 global fire emissions, using emission factors from Andreae et al (ACP, 2019), but replacing EFs for certain biomes with mean EFs from Friedli et al, (2003a;b) and McLagan et al, (2021). They also apply those EFs to GFED4 fire emissions for sensitivity simulations. Those biome-specific EFs could be applied 570 to the chosen fire emissions dataset (GFASv1.2) for this project to generate consistent Hg fire emissions.

Similarly, EFs used for POPs and PAHs could be added and applied to the base fire (and anthropogenic) emissions for this study. For PAHs, there is a recently updated Peking University (PKU-FUEL) "global PAH emission inventory" spanning from 1961 to 2020 at http://inventory.pku.edu.cn/, which takes wildfire emission into account, and used measured PAH emission factors. That group also developed global OPFR, SCCP, and PCDD/Fs emissions, often using experimentally 575 derived emission factors from the USEPA and UNEP as well as the literature (He et al, 2004; Jiang et al, 2017; Song et al, 2022;2023; Li et al., 2023).

*Post-fire dust emissions*

The removal of vegetation creates a more exposed soil surface from which dust can be emitted (Dukes et al., 2018; Jeanneau et al., 2019, Whicker et al., 2006). The emission of dust from a post-burn landscape will continue until the vegetation





sufficiently recovers, spanning a period of days to potentially years. Approximately 1-in-2 large fires are estimated to be followed by increased dust emissions, with savanna ecosystems the most susceptible (Yu and Ginoux., 2022). Emission estimates are highly uncertain with the only global estimate to date of 100 Tg/year of additional soil dust emissions with an order of magnitude uncertainty (Hamilton et al., 2022). As there are no existing emissions datasets for this process, further research beyond the scope of this study would be needed to address this impact of fires.

*Wildland urban interface fires*

Wildland Urban Interface (WUI) fires account for ~4% of total fires globally. WUI fires can involve built-structure burning, and hence their emissions may be more harmful. They are also closer to humans and properties causing expensive damages. Studies have been conducted for specific WUI fires and regions (Holder et al., 2023) and future version of FINN will include WUI fire emissions. However, currently there is no global BB emissions dataset that explicitly addresses WUI fire emissions

and future research beyond the scope of this study would be needed to address this aspect of fire emissions.

*Summary of recommended historical fire emissions dataset*

The discussions at FEW 2023 suggested that several characteristics are important in selecting a fire emissions dataset for the multi-model experiments: 1) high temporal resolution, given the high variability of fires; 2) the inclusion of boreal peatland fires, particularly for those interested in boreal and Arctic locations; and 3) the inclusion of fire plume height for atmospheric

modelling. For these reasons, GFASv1.2 became the recommended fire emissions dataset for the historical period (starting in 2003), with updated fire type map, and emission factors (thus, GFASv1.2+).

## 4.2.3 Future biomass burning emissions

Land models, such as those that participated in FireMIP (Li et al, 2019) can provide fire emission projections that are reflective, not only of future land use changes, but also of the changing climate under different future climate scenarios.

However, as of this writing, the FireMIP future simulations have yet to be conducted.

Current CMIP6 SSP future fire datasets only account for human impacts on future fire activity, whereby fire activity is assumed to decrease and includes no impact from the changing climate conditions on those future fire emissions (Figure 2, left bars). An alternative set of future fire emission projections does however exist in six fire-climate-coupled models from CMIP6 (Xie et al., 2022). The models have future fire results that take into account the changing climate under different SSP

scenarios: "a climate-consistent future fire emissions estimate" (Figure 2, right bars). Emissions for three SSP scenarios (SSP1-2.6, SSP3.70, SSP5-.8.5) have been produced by Hamilton and Kasoar et al. (submitted), and other SSP scenarios using the same methodology can be generated (e.g., SSP2-4.5 for this study) for 2015-2100. In each emission projection "natural" fire emissions are defined as boreal and temperate forest and all grassland fires and calculated as a product of the CMIP6 multi-model mean, accounting for similarities in land models. "Human" controlled agricultural and deforestation fires are then added to natural fires from the SSP dataset. However, those same agricultural fires are included in the GAINS

anthropogenic emissions (Section 4.4.1) and shouldn't be double-counted (see Section 5.2 for this recommendation). Tropical forest fires are assumed to be primarily due to deforestation practises and were also added from the SSP dataset in





place of CMIP6 model estimates in that biome. Peat fires are held at present day levels throughout the century, very likely underestimating their contribution to future emission fluxes, but this is because the interactive ESM fire modules did not

contain these uncertain types of fires. Finally, each emission dataset is bias corrected regionally to emissions in the present day, currently this is GFED4s but could be corrected to GFASv1.2.

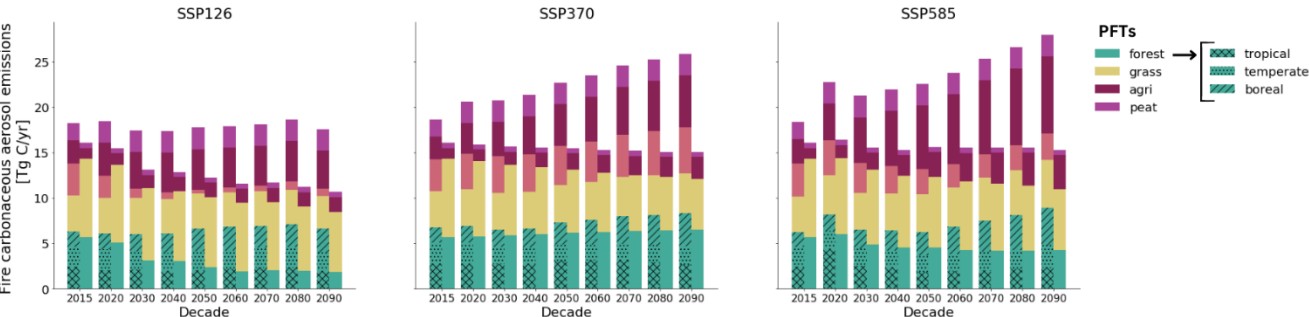

**Figure 2**: Decadal average timeseries of future fire aerosol emissions from Hamilton et al (submitted). Right-hand bars are fire emissions from interactive ESMs and left-land bars are fire emissions from CMIP6, based on land use change only.


Note that in Tang et al (2023) the Community Earth System Model 2 (CESM2) was used to project future burned area and total fire carbon emissions under different climate scenarios, however, it is only based on the one ESM, so it is not recommended for this project. As mentioned in Sections 2.4 and 3.2.4, applying fire management policies to future scenarios is beyond the scope of this study, however, further information on that topic is included in the Supplement, Section S.3.3.

**4.2.4 Other emissions**

Aside from the emissions mentioned above, models typically include biogenic and geological emissions from natural sources. These can include isoprene and other VOC emissions from vegetation, $NO_x$ emissions from soil microbes and lightning, sulfur emissions from volcanos, etc. Most models rely on the same interactive biogenic emission database, namely MEGAN (available at https://bai.ess.uci.edu/megan; Guenther et al., 2012) (Table A.2), or a derivative thereof.

**4.3 Available meteorological inputs**

The height reached by a smoke plume, its horizontal transport, vertical mixing, and subsequent impact on a region are greatly determined by the prevailing weather conditions. These effects occur across a wide range of scales, from turbulent mixing of pollutants in the boundary layer, lifting into the free troposphere, and subsequent transport by the prevailing winds. For models not generating their own meteorology, observation-based reanalysis datasets provide an important source

of meteorological information needed to drive the models (included in Table A.2), although differences between available products provides an additional source of uncertainty (e.g., Adebiyi et al., 2024).





### 4.3.1 Historical meteorological datasets

Currently, several meteorological reanalysis data sets are available and could be utilized, such as MERRA2 (Modern-Era Retrospective Analysis for Research and Applications, version 2), ERA5, NCEP-NCAR (National Centers for Environmental Prediction - National Center for Atmospheric Research), or JRA-55 (Japanese 55-year Reanalysis), among others. They are summarized in Table S2 in the Supplement.

### 4.3.2 Future meteorological input

To assess the alterations in meteorological conditions across the 21$^{st}$ century and their potential implications on fires (frequency, intensity, transport), the meteorological datasets provided by the Shared Socioeconomic Pathways (SSPs) climate projections can be utilized. The IPCC (Intergovernmental Panel on Climate Change) defined the SSPs scenarios which illustrate different potential pathways for societal development throughout the 21$^{st}$ century and analyse their potential impacts on greenhouse gas emissions.

The SSPs are classified into five trajectories: SSP1 represents a sustainable world, SSP2 outlines a moderate pathway, SSP3 depicts a fragmented world with considerable challenges, SSP4 illustrates a world emphasizing equality and sustainability, and SSP5 envisions a world driven by rapid economic growth and dependence on fossil fuels.

These five categories define different SSP emissions and concentration pathways, providing unprecedented detail of input data for climate model simulations: SSP1 (1.9 and 2.6), SSP2-4.5, SSP3-7.0, SSP4 (3.4 and 6.0), and SSP5 (3.4-OS and 8.5). The SSPX-Y scenarios refer to the estimated RF levels at the end of the 21$^{st}$ century; for instance, the '1.9' in the SSP1-1.9 scenario signifies an estimated RF level of 1.9 W m$^{-2}$ in 2100. They provide a more in-depth analysis of climate drivers and responses than the previous RCPs (Representative Concentration Pathways) employed in AR5 (Chuwah et al., 2013).

The SSPs are derived from model simulations conducted under the Coupled Model Intercomparison Project Phase 6 (CMIP6) (Eyring et al., 2016). Access to the meteorological fields generated by the global climate models (GCM) under each of the specified SSP scenarios, is facilitated through platforms such as those provided by CMIP6 (https://pcmdi.llnl.gov/CMIP6/), the IPCC Data Distribution Centre (DDC) (https://www.ipcc-data.org/), and the Climate Data Store (CDS) by Copernicus (https://cds.climate.copernicus.eu/), among others.

### 4.4 Observational data available for model evaluation

The comparison of model results to observations is valuable for assessing how well models represent the real world and is critical for identifying gaps in our current understanding or weaknesses in how key processes are represented in models. Given the known uncertainties in fire and other model processes, observational comparisons provide a valuable opportunity to critically assess current parameterizations and identify which are most appropriate under particular conditions.

Comparisons with satellite observations of atmospheric composition will enable large-scale simulations to be evaluated consistently over the historical period under consideration. Observations of CO and aerosol optical thickness will be used to





evaluate long-range transport simulation (transport pathways, plume dilution). LIDAR observations will be used to assess the transport altitude and vertical extent of plumes. All surface monitoring measurements of the pollutants of Section 3.1
could be used for model evaluation, but we focus the rest of this section on highly-relevant fire-specific observational datasets and field campaigns (see also Table S1 in the Supplement for the list of suggestion observational datasets).

As shown in Section 3.1, there is no single tracer which is emitted by wildfires only, and domestic wood burning has the same signature as wildfires. Enhanced Hg and POP concentrations are also often observed in atmospheric concentrations of biomass burning emission and come from the burned matter itself, as well as reemitted from soil. If those substances have
co-located enhancement with other primary pollutants like CO, BC, and SOA, it is a strong indication that wildfire emissions are observed (e.g. Eckhardt et al., 2006).

For detecting wildfire plumes in observations, statistical methods use a combination of different trace species. For example, $SO_4$, BC, CO, $NO_2$ have been combined with a positive matrix factorization to identify biomass burning plumes (Karl et al., 2019). Yttri et al., (2023) used aerosol absorption coefficients recorded at different wavelengths by an aetholometer to
distinguish BC emitted by fossil fuel or by biomass burning. Those observations are available for several stations in Europe.

Evaluation of modelled fluxes (e.g. deposition) are more challenging. These, as well as Nr impacts, may also be dynamically modelled in some systems. Cross-disciplinary satellite (atmospheric, land cover, water quality, etc) and in-situ data can be used to evaluate modelled deposition results, helping identify weakness in individual models and reduce uncertainty in impact assessments (e.g., Fu et al., 2022).

For the remote Arctic, the highly variable, non-changing long-term time trends of PAHs are inconsistent with the global PAH emission reduction and have significantly increased during summers with more frequent wildland fire events in Nordic countries (Yu et al., 2019). Retene (a PAH) was often used as tracer for wildland fire activities. However, volcanic eruption (Overmeiren et al., 2024) and volatilization from soil and ocean due to warming can also elevate PAHs' air concentrations in remote locations. Models together with observations can better link BB and long-range transport of fire-related substances to
remote sites.

## 4.5 Experiment design and sensitivity analyses

This section outlines different model experiments to help answer the science policy questions of Section 2. These fall into several distinct sets, targeting different aspects of our understanding, and some include a range of sub-experiments to explore specific aspects in greater depth. Model groups may contribute to any number of experiments but are not required to
complete them all. Where applicable, we indicate in Section 5.4 which experiments are higher priority for HTAP, and which experiments may be dependent upon completion of other experiments.

### 4.5.1 How well do models perform? Baseline and case study simulations

Models should conduct baseline simulations of recent historical conditions, with a common set of anthropogenic and fire emissions, as both a basis of comparison for perturbation and sensitivity experiments and for general model





intercomparisons and evaluation with observations. The results can then be used to quantify the uncertainties and variability in atmospheric modelling. As the type of models participating is highly variable, with a range of computational costs, both short and long time periods are suggested for the baseline simulations. These time periods will be selected based on the availability of reliable emission assessments and periods with abundant observations. Very computationally expensive models (e.g. very high resolution, inclusion of complex atmosphere chemistry) may only be able to simulate one year or less.

Given how highly regional and interannually variable fires are, we can identify short-term fire case studies for evaluation of those models and explore particular fire events in detail. Fire event case studies may include the particularly large Australian fires of 2019-2020 (Filkov et al. 2020; Johnston et al., 2021; Collins et al., 2021; van der Velde et al., 2021, Anema et al., 2024); the fires in the U.S that coincided with the 2018 WE-CAN and 2019 FIREX-AQ measurement campaigns (Juncosa Calahorrano 2021; Warneke et al., 2023); and the significant fire season in Indonesia in 2015 due to a strong El Niño (Chen

et al, 2016; Nechita-Banda et al., 2018).

### 4.5.2 What is the magnitude of pollution that comes from fires? Source-receptor/emissions perturbation experiments

To determine the magnitude of pollution from fires, species concentrations from baseline simulations can be compared to simulations with fire emissions removed. For additional detail, fire emissions from different geographical regions and from different types of burning can be perturbed for separate species, locations, and seasons to quantify source/receptor

relationships and their uncertainties. However, the number of perturbation experiments can increase rapidly, so care is needed to prioritize and not define regions and sectors too finely.

*Geographical Regions*

Coarsely-defined regions help reduce the number of perturbation simulations. Figure 3 shows several options for the geographical source regions, including those used within the HTAP2 multi-model experiments (Fig 3a). These distinguished

boreal fires in higher latitudes from the low-latitude fires associated with agricultural, temperate and grasslands. These source regions should be further refined, particularly in the eastern hemisphere, where we could separate Europe from Asia in the NE box, and separate Africa and the Middle East from SE Asia in the SE box. Regions used for anthropogenic emissions perturbation experiments in the HTAP3 OPNS project are shown in Fig 3b. We note that the southern hemisphere Africa, has been a focus of recent field campaigns (Zuidema et al., 2016) as the region emanates a third of the world's carbon

from biomass burning aerosol (van der Werf et al., 2010). South America also emits significant fraction of the world's total BB aerosols. Therefore, for global modelling completeness, the scientific modelling community would benefit from including those southern hemispheric regions in the perturbation experiments as well. Figure 3c shows the GFED BB emissions regions used in many analyses that balance political regions and fire-relevant biomes. However, there are 14 GFED regions, and we merge these into 8 regions to make perturbation experiments (exp 5 in Table 5) more feasible in

Figure 3d. These merged regions are broadly consistent with the HTAP2 regions, but with improved coverage, and are loosely aligned with the regions used for anthropogenic emissions in HTAP3. Regional models may have geographical domains that differ from these, and where possible, these should simulate a subset of the regional perturbation experiments.





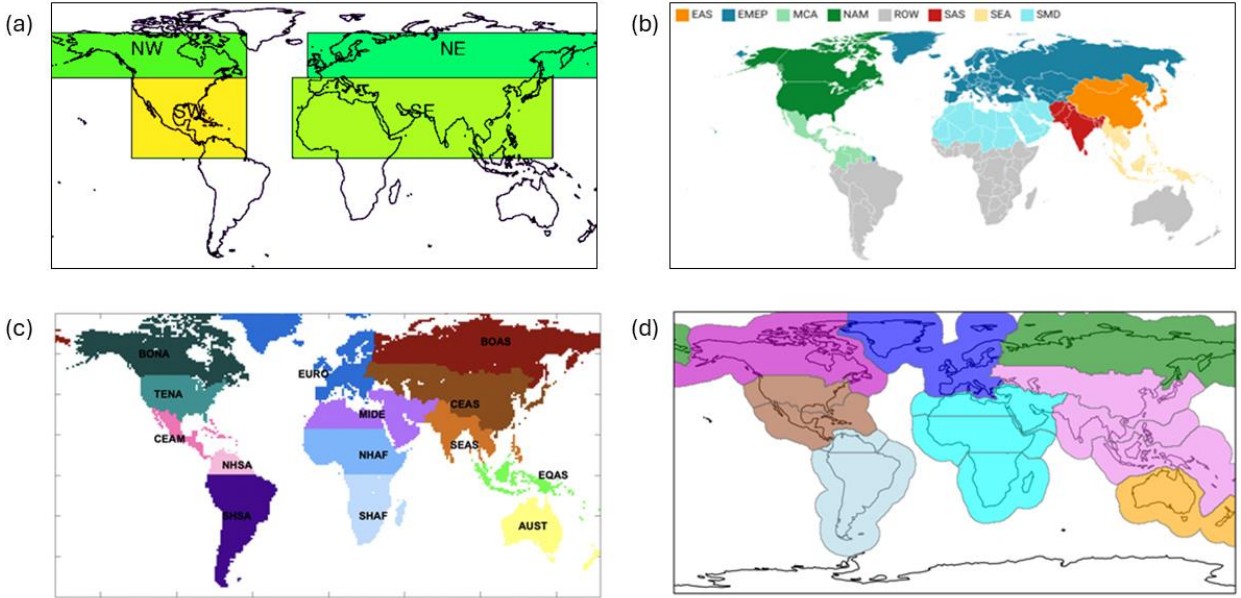

**Figure 3:** Possible regions for perturbation experiments. (a) BB source regions used in HTAP2 experiments. (b) regions
used for anthropogenic emissions in HTAP3 O3PNS project, (c) GFED regions often used for fire emissions datasets, and
(d) GFED regions (grey lines) and proposed merged regions (coloured areas).

*Fire Sectors*

Management decisions and policies are best informed by perturbing biomass burning sectors separately. The two main
categories are agricultural burning and wildland fires. Agricultural biomass burning is the deliberate burning of agricultural
waste products, such as crop waste products, stubble, and other organic matter left in fields after harvest, as a method of
waste disposal or as a practise in land management. The burning of grasslands towards coaxing new growth is also included.
Deliberate burning is frequently applied in agricultural areas, especially where traditional practises are still widely practised.
The United National Economic Commission for Europe (UNECE) adopted a guidance document on how to define and build
policies around reducing open agricultural burning (UNECE, 2022).

Perturbing emissions from these two sectors separately over the 9 regions (8 regions + all) implies that global models
participating in perturbation experiments would have 18 simulations to run. Figure 4 shows the distribution of the dominant
fire types in GFED3 (adapted with peat and soil organic matter from Kaiser et al. 2012). An extrapolation that covers all land
is used in GFASv1.2 (and v1.4) to derive and apply fire type-specific fire radiative power (FRP)-to-dry matter burnt
conversion factors and to apply fire type-specific emission factors for the different smoke constituents.



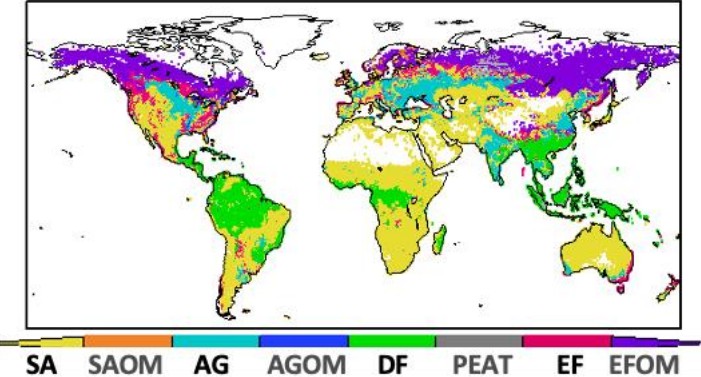

**Figure 4**: Fire types adapted from Kaiser et al. (2012): SA=savannah fires; SAOM=SA with potential soil OM burning;
AG=agricultural fires; AGOM=AG with potential soil OM burning; DF=tropical fires; PEAT=peat burning; EF=extra-
tropical fires; EFOM=EF with potential soil OM burning.

### 4.5.3 What is the impact of different fire processes? Process perturbation experiments

Much of the uncertainty in the wider impact of fires arises from weakness in our understanding of fire processes, in their
representation in models, and in the sensitivity of the impacts are to these treatments. These uncertainties can be explored
with simple sensitivity studies perturbing key processes one at a time. Key uncertainties include the magnitude and timing of
emissions, the plume rise associated with the heat of the fire, the meteorological conditions, and the chemical and deposition
processes occurring during plume transport.

*Fire plume height*

Most of the pollutants emitted from wildfires are released directly into the atmospheric boundary layer. However, depending
on the meteorological conditions and the strength of the fire, material can be lifted well into the free troposphere or in
extreme cases, the stratosphere. This can have a substantial influence on the downwind impacts of the fire, as horizontal
transport is typically faster in the free troposphere, the chemical processing of oxidants such as $O_3$ is typically more efficient,
and the removal of pollutants by wet and dry deposition processes is less efficient than in the boundary layer. Previous
model studies have quantified the importance of injection height for key pollutants (e.g., Leung et al., 2007; Feng et al, 2023)
but this has not been explored in a rigorous manner across a range of models. On longer timescales, the presence of high
levels of BC in plumes can lead to local heating which causes further lofting of the plume (e.g., Ohneiser et al., 2023). The
altitude of tropospheric O3 also influences the magnitude of its warming potential. Therefore, fire plume height introduces
substantial uncertainty into assessment of impacts.

Some fire emission datasets (such as GFAS and GBBEPx) are based on FRP, whereas, others, like GFED and FINN
emissions, are based on burned area (BA). Both FRP and BA are (mainly) based on MODIS satellite observations. Daily
information on wildfire injection heights, and/or FRP (Fire Radiative Power), in combination with meteorological
parameters, can be used in the calculation of injection heights. Daily fire emissions based on FRP and BA differ substantially



on a daily basis. Some fire emissions datasets, such as GFAS provide injection height parameters based on satellite-observed FRP and available meteorological parameters (Remy et al., 2017).

Some models represent plume rise in their simulations while other models do not. Among these models that address plume rise, some include online parameterization of fire plume rise. E.g., the Freitas scheme (Freitas et al., 2007; 2010) which calculates plume rise by solving a set of 1-D differential equations vertically.; the Sofiev scheme which considers the conservation of the heat energy (Sofiev et al., 2012; 2013); and the Canadian Forest Fire Emission Prediction System (CFFEPS), which contains a thermodynamically-based fire plume height parameterization based on fire energy and neutral

buoyancy (Chen et al., 2019). Other models use a simpler approach of a constant plume injection height climatology (e.g., Dentener et al, 2006; Val Martin et al., 2010; 2012), which usually depends on region, season, and vegetation type and does not consider FRP or fire size for specific fires. It is important to understand the impacts of different plume rise treatments on the model results, exploring the impacts of fuel type, moisture and heat flux assumptions across the plume rise schemes used. Data from CALIPSO, MISR, TROP-OMI (Griffin et al., 2020), and airborne instruments (Shinozuka et al., 2020; Doherty et

al., 2022) would be useful for model evaluation of the effects of plume rise.

Impacts of fire plume height were found to be different when looking at regional simulations versus at global climatological scales; in Field et al (2024), using GFAS injection heights in the model was important for improving model performance at regional scales, whereas it was long-range transport patterns, influenced by the winds in the driving meteorology, that mattered more than individual fire events at climatological time scales.

*Fire plume chemistry*

Biomass burning emits particles along with $NO_x$, nitrous acid (HONO), ammonia ($NH_3$), CO and $CH_4$, and hundreds of VOCs, including a large number of oxygenated VOCs (OVOCs) (Jaffe et al., 2020). Representing this chemical complexity is a key challenge for modelling fire impacts on air quality, especially for secondary pollutants such as $O_3$ (Section 3.1.1) and SOA.

State-of-the-art atmospheric chemistry models typically overpredict $O_3$ close to fires but have difficulty simulating the influence of aged wildfire smoke plumes on downwind $O_3$ (e.g., Pfister et al., 2008; Singh et al., 2012; Zhang et al., 2014; Fiore et al., 2014; Lin et al., 2017; Baker et al., 2016; Zhang et al, 2020; Jaffe et al., 2020). This may reflect: (1) inaccurate fire emissions, especially underestimates of oxygenated VOC emissions from wildland fires (Arnold et al., 2015; Jin et al., 2023; Permar et al., 2023; Lin et al., 2024b); (2) lack of sufficient resolution or parameterization of smoke plume rise

dynamics (Paugam et al., 2016; Ye et al., 2023); (3) shortcomings in model representation of rapid photochemical processes in a concentrated smoke plume (Singh et al., 2012). Several modelling studies have shown strong sensitivity of $O_3$ production to differences in VOC chemistry, fire plume vertical transport, and NOy partitioning (Zhang et al., 2014; Arnold et al., 2015; Lin et al., 2024b). Rapid conversion of $NO_x$ to more oxidized forms typically reduces excessive ozone production simulated in near-fire smoke plumes. A recent study by Lin et al. (2024b) shows that sequestration of wildfire

NOx emissions in Canada as PAN enhances ozone production during smoke transport and thereby increases the impacts of Canadian wildfires on ozone air quality in US cities.





Additionally, large uncertainties in carbonaceous aerosol emissions from biomass burning (Pan et al., 2020; Carter et al., 2020; Xie et al., 2020) can also influence simulations through the impacts of aerosols on heterogeneous chemistry and photolysis rates. Further suggestions for model experiments to assess $O_3$ chemistry uncertainties appear in Section 5.3.1.

An additional challenge is the rate of SOA formation (Section 3.1.3). While SOA formation increases near-source, measurements taken after long-range transport suggest SOA loss (Sedlacek et al., 2022; Dobracki et al., 2024), hypothesized to occur through heterogeneous oxidation primarily. Estimates of the reaction rates with OH vary, and measurements focused on constraining these rates would improve model depictions. OA loss is not included in many models (e.g., Lou et al. 2022 only considers photolysis, although their modelling construction could be using photolysis as a proxy for

heterogeneous oxidation OA loss as well) but could be encouraged in the model output for this project.

### *Dry and Wet deposition*

Modelled dry and wet deposition fluxes are highly variable, uncertain, and a possibly significant cause for inter-model differences in pollutant concentrations. Models can test out different wet and dry depositions schemes, and/or to turn deposition on and off to quantify its impact. Deposition is also important for evaluating ecosystem impacts. Wet and dry

deposition fluxes should be diagnosed from all model simulations.

### 4.5.4 How will fires and their impacts change in the future? Future scenario experiments.

The frequency and severity of wildland fires are likely to increase within a warming climate, particularly in the northern hemisphere (van Wees et al., 2021). Quantifying the influence of these changes, given different future emission scenarios, is an important application of models (e.g., Xie et al., 2022). Future modelling experiments can be formed with chemical

transport models provided future emissions and meteorology are provided. Experimentation can also be performed by ESMs with and without interactive fire modules. ESMs can typically simulate future climate/meteorological conditions in a free running state out to 2100. Experiments for future fires with both interactive ESMs and other atmospheric models driven offline will help determine the range of uncertainty on future fire projections and their impacts. While fire emissions are likely to change under the effects of changes in human management practices and policies, those aspects that aren't already

included in the CMIP SSP scenarios (Xie et al., 2022) are beyond the scope of this study due to a lack of scenario emissions datasets.

## 5. Recommended Plan

### 5.1 Simulated time periods

Given the combination of emissions dataset availability (Section 4.4) and existing observational datasets to compare against

(Section 4.4), we suggest the following time frames for simulation years (Table 3). The short historical option for the HTAP3 OPNS and Hg projects was selected to be 2015. However, 2015 had a strong El Niño and was an extreme fire year in Indonesia as a result. Fires are so greatly variable on interannual scales, that it would be unwise to base policy decisions





on analysis of a single year. We therefore encourage use of the medium historical option. which includes field campaigns of 2016-2018 that were offshore from African fires (Redemann et al., 2021; Haywood et al., 2020; Zuidema et al., 2018), and
2019 which had a field campaign in the US. The medium option stops by the end of 2019 to avoid incorporating the complexity in anthropogenic emissions that arose with the COVID-19 pandemic in 2020. The medium future option includes 5 years on either side of the 2015 start and 2050 end dates of the GAINS future emissions to enable 10-year averages to be created around these start and end dates, thus accounting for interannual variability, (consistent with the HTAP3 OPNS project). The 2015 emissions may be used for 2010-2014, and the 2050 emissions for 2051-2055). Finally, while the climate
community routinely does simulations out to 2100, given that the GAINS anthropogenic emissions end in 2050, and the AerChemMIP2/CMIP7 community (see Table A.1 and Section 3.3) will focus on future simulations, including climate impacts from fires, we have elected not to include a long future option within this study.

**Table 3:** Simulation time periods, with options for different types of models.

|  | Short option | Medium option | Long option |
|---|---|---|---|
| **Historical** | See Case studies (Sec 4.5.1) | 2015-2019 | 2003-2020 |
| **Future** | 2050 | 2015-2050 | N/A (see Section 3.3) |


### 5.2 Inputs: Emissions and Meteorology

Based on discussions in Section 4.4, the following emissions datasets are recommended, and summarized in Table 4 below.

*Historical Fire emissions:*

The historical fire emissions datasets were carefully considered during and following a 4-day online workshop hosted by
IGAC BBURNED in November 2023. The methodology, advantages, and disadvantages of each major global fire emissions dataset were discussed. It was agreed to recommend use of the GFASv1.2 fire emissions because (a) they provide daily emissions (providing improved temporal variability over monthly emissions), (b) they include peatland fires, including in the boreal region, the latter particularly important for the AMAP scientific community, and (c) they provide fire plume heights as well as speciated emissions. We note, however, that peat fire emissions remain highly uncertain, and that these fire
emissions do not include special treatment of WUI fires.

*Future fire emissions:*

The future fire emissions dataset that is derived from a multi-model ensemble that includes the influence of the changing climate on fires is that from Hamilton and Kasoar (personal communication; Hamilton, Kasoar, et al, submitted). We recommend use of this future fire emissions dataset, but note that it does not include scenarios for future changes to fire
management policy and practice, as these quantitative emissions adjustments are not available yet.

*Historical anthropogenic emissions:*





These are chosen to be consistent with the other concurrent HTAP3 projects. They are the HTAP v3.1 anthropogenic emissions (expected to be delivered in September 2024), and include all relevant species (Section 3.1), over 2000 to 2020, inclusive.

*Future anthropogenic emissions:*

For consistency with other HTAP3 projects, the CLE (current legislation) future emissions from IIASA GAINS (LRTAP) will be used in future simulations. Climate modellers may wish to simulate out to 2100, and while the SSP2-4.5 anthropogenic emissions for 2015 to 2100 are available and are roughly equivalent to the GAINS CLE emissions scenario for $CO_2$ and energy, they are not necessarily similar for other pollutant emissions. We therefore recommend for this project

ending the future simulation in 2050 and participating in CMIP7/AerChemMIP2 for longer future simulations.

*Biogenic and other natural emissions:*

While it is useful to have consistent emissions across models, this can be difficult to achieve due to the dependence of natural emissions on structural aspects of models including vegetation, soils and land use. Therefore, we suggest that each modelling centre use their preferred emissions from biogenic and other natural sources. These should be documented and

taken into consideration in the analysis.

**Table 4:** Emissions Inputs for model experiments

| Emission type | Recommendation | Notes/references | Download location (if currently available) |
|---|---|---|---|
| **Historical Fire** | GFASv1.2+ for BB, including agricultural burning (2003-2024) | Daily gridded global 0.1-degree resolution. Including its agricultural burning emissions.<br><br>Note: these will be updated in the near future to include newer emission factors | https://ads.atmosphere.copernicus.eu/cdsapp#!/dataset/cams-global-fire-emissions-gfas?tab=overview |
| **Future Fire** | Hamilton, Kasoar et al (2015-2100) | SSP2-4.5-scenario-based climate-influenced future fire emissions, calibrated to GFASv1.2 historical fire emissions. Minus agricultural burning. | TBD |
| **Historical anthropogenic** | HTAPv3.1 (2000-2020) | Minus agricultural burning<br><br>Note: These will also be updated by the end of September 2024 to include 2019 and 2020 (v3.1) as well as updated emissions for 2000-2018. | v3 for 2000-2018 https://edgar.jrc.ec.europa.eu/dataset_htap_v3, v3.1 TBD |
| **Future anthropogenic** | IIASA GAINS CLE (1990-2050) | Including agricultural burning | https://zenodo.org/records/10366132 |
| **Biogenic and other natural emissions** | MEGAN, or each modelling centre use their default | MEGAN or models' own | N/A |





*Driving meteorology*

As discussed in Section 4.3, there are several data reanalysis collections that could potentially be employed. Although the ERA5 collection offers greater spatial, temporal, and vertical resolution overall, any of the mentioned datasets would be suitable for use. It is recommended that modellers use ERA5 if possible, but otherwise use their preferred meteorology for historical simulations and ensure that they document this clearly. For future simulations, we suggest using interannually varying, monthly mean sea surface temperatures and sea ice distributions from SSP2-4.5.

**5.3 Model experiments**

The following model experiments in Table 5 are proposed based on the discussions in Section 4.5, and further details for selected experiments is described below.

**Table 5:** Model experiment (exp) types

| Exp name | Description | Purpose | Priority |
|---|---|---|---|
| 1. Baseline simulation | Historical time period(s) given in Table 3. Common set of emissions given in Table 4. | Model evaluation; baseline for subsequent sensitivity exps | High |
| 2. Case study(ies) | More detailed, specific fire events at higher spatial and temporal resolution | Model evaluation | High for regional models. Low for global models. |
| 3. Fire emissions sensitivity | Same as exp 1, but driven by different sets of fire emissions (GFED, FINN, etc) | Model/emissions evaluation; to gauge differences between fire emission datasets across models | Low |
| 4. Prescribed future fires | Future time period(s) given in Table 3. Future emissions given in Table 4. | To determine how wildland fires and their impacts will change in the future | High |
| 5. Regional and sectoral emissions perturbations | Turn off all BB emissions for all species everywhere.<br><br>Turn BB emissions off in each region of Figure 3d, and each of the 2 sectors: agricultural burning and wildland fires, over the historical time periods in Table 3. | To quantify regional source/receptor relationships and uncertainties | High<br><br>Priority: Both fire sectors together; separate sectors if resources permit. |
| 6. Fire process | Parameter/process perturbations, for | To determine importance of | Medium |





| perturbations | fire plume height, chemistry, emissions, and meteorology (see Section 5.3.1). Short-to-middle time periods of Table 3. | different processes and impacts of different model fire parameterizations | |
| --- | --- | --- | --- |
| 7. Interactive fire modules | Historical and future simulations (Table 3) with coupled land-atmosphere models. | To determine how wildland fires will change in the future with an interactive climate and compare to exp 4 results. | Medium |
| 8. Data assimilation | Inverse modelling to combine CTMs with observed atmospheric VMRs | Infer surface-atmosphere emissions/fluxes | Low |


### 5.3.1 Details for fire process perturbation experiments (exp6)

While a short time range for perturbation experiments can help keep model simulations manageable, they may not provide generalizable results, given the high interannual variability of fires. Therefore, the time ranges of Table 3 should be followed for perturbation experiments as well.

*Injection height:* Repeat of exp1 with alternative fire plume height definitions. We suggest the following options, where modellers can opt into any number of these when possible:

- model's default fire plume height system, whatever it may be,
- the plume heights provided by the baseline fire emissions dataset (GFASv1.2), that are FRP-based (Section 4.5)
- climatological plume rise from AEROCOM (Dentener et al., 2006), assuming standard vertical profiles, and
- no plume rise: assuming all pollutants are released into the lower part of the planetary boundary layer.

*Chemistry:* To assess the impacts and uncertainties around fire plume chemistry, a few sensitivity runs are recommended:

- Partition total $NO_y$ emissions from biomass burning into PAN (37%), $HNO_3$ (27%), and $NO_x$ (36%), rather than emitting only NO in the baseline simulation (exp1), as recommended by Lin et al. (2024a, 2024b) based on recent aircraft measurements (WE-CAN 2018 and FIREX-2019).
- Doubling BB emissions of all NMVOCs, including formaldehyde and acetaldehyde producing acetyl peroxy radical (CH3CO3) for PAN formation;
- Increasing BB emissions of OC and BC aerosols by 50% to explore their impacts on oxidative chemistry through heterogeneous chemistry or photolysis.

*Emissions temporal resolution:* Repeat of exp1 with hourly, daily, and monthly versions of the fire emissions to quantify the importance of temporal resolution. Many previous major studies, such as CMIP6, have used monthly fire emissions and this sensitivity study will allow these results to be placed in context.





*Meteorology*: Use repeating annual meteorology for 2018 with interannually changing emissions to determine how much of
interannual variability in impacts seen in exp 1 is due to meteorology, and not emissions.

### 5.3.2 Details for future experiments (exp4)

The SSP2-4.5 future climate scenario will be the driver for the future time period, which includes those future fire emissions
from Hamilton & Kasoar (submitted), and the GAINS CLE anthropogenic emissions. For future agricultural burning
emissions, which appear in both the BB and the anthropogenic emissions datasets, we recommend that the GAINS future
agricultural burning emissions be used, and those removed from the BB emissions so as not to double-count them.

### 5.4 Model outputs

To maximise accessibility of the results, the model output data request for this project is based on the AerChemMIP tables
from CMIP6, as adopted by the HTAP OPNS project, with some additions for the extra species and impacts. The Table of
model outputs is located online at https://nextcloud.gfz-potsdam.de/s/sp8XmMY2rQizjA4. We have added Hg, POPs, and
PAHs, and place greater priority on hourly surface $NO_2$, $PM_{2.5}$, and $O_3$, as well as hourly $O_3$ deposition parameters needed
for air quality, health, and ecosystem impacts analysis, in addition to monthly radiative forcing output for climate impacts.
When measurements and impacts are only related to surface concentrations (e.g., POPs), we have suggested only surface-
level 2D model output be provided to save storage space.

*Data workspace*

The model output can be uploaded to METNO's AeroCom database and infrastructure as part of the HTAP3 component of
the AeroCom database. Instructions for obtaining access to the aerocom-user server, formatting, uploading, downloading are
found here: https://aerocom.met.no/FAQ/data_access. The AeroCom database infrastructure is available to host HTAP
model data on a read-only permanent database, which can be accessed by authorised users with an account on the aerocom-
user server. A scratch area on the AeroCom-user server can be used to upload data. Uploaded data can be transferred on
demand by METNO to the read-only permanent database section for HTAP, under the directory HTAP-PHASE-III.

### 5.5 Post processing and analysis

This multi-pollutant, multi-model experiment will generate a large amount of data that will be analysed to answer the science
policy questions of Section 2.

### 5.5.1 Model evaluation: Comparison of experiments 1, 2, 3, and 6 to observations

By comparing the results of experiments 1, 2, 3, and 6 to the observations discussed in Section 4.4 (and listed in Table S0),
specific model inputs and processes can be evaluated. The aim of the evaluation would be to improve our understanding of
fire processes, such as plume rise, plume chemistry and improve their parameterizations in models. We may also be able to





determine which inputs (emissions, meteorology) and parameterizations are best, as well as identify gaps that require further

research. One example would be to analyse the impacts of injection heights on PAN concentrations in the free troposphere and downwind $O_3$ formation as fire plumes subside into the boundary layer, by comparing the model simulations of PAN and related tracers to recent aircraft measurements. We suggest that, when possible, community tools like MELODIES (Model EvaLuation using Observations, DIagnostics and Experiments Software), and ESMValTool be used for inter-model comparisons and evaluation against observations. Regardless, the evaluation will require a large effort by the community.

**5.5.2 Assessing health impacts of fires**

The most cited and widely used approaches of risk analysis are: all cause of deaths; mortality and morbidity impacts; emergency hospitalization; reduced life expectancy; premature mortality; incremental life-time cancer risk; and health-related cost of air pollution (Goel et al., 2021; Sonwani et al., 2022; Nagpure et al., 2014; Gidhagen et al., 2009; Guttikunda et al., 2014; Ghozikali et al., 2014; Farzaneh, 2019). Human health risk assessment is the mathematical estimation and

modelling of several processes, including population estimates, population exposure to pollutants, and adverse health impacts assessment through specific concentration-response functions (WHO, 2021). Widely used quantitative health risk assessment tools of different agencies have been listed in Table S1. While Table S2 represents the comparison between the air pollution health risk assessment tools (methodologies, scopes, input parameters, and predicted health impacts). The surface-level model outputs of atmospheric composition at high spatial and temporal resolution will be invaluable for new

health risk assessments.

**5.5.3 Assessing climate impacts of fires for historical and future time periods**

Climate impacts can be assessed through the RF from fire-emitted pollutants by comparing the RF with and without fire emissions. These should be included in the regional perturbation experiments in order to have the required data to assess the RF impacts of biomass burning. For estimation of future fire RF, it is key to quantify the effects caused by the non-linearities on the $O_3$ RF and for the aerosol-cloud interactions. Source attribution techniques for $O_3$ were used in Grewe et al. (2017)

and Butler et al. (2018), and for aerosols in Righi et al. (2021).

**6  Conclusions**

In this paper we have described the need for a multi-model, multi-pollutant study focused on fires, and highlighted a range of important science-policy questions arising from discussions with the scientific and policy communities that this study is

intended to answer. The study will address gaps in our current scientific understanding of fire processes and provide a more robust quantification of fire pollution and its impacts to inform decision-making. We have thoroughly discussed the scope of this study, based on extended consultation with the science, impacts and policy communities, and have outlined a number of alternative design options. We then provide recommended specifications for a modelling study to be carried out over the



next ~3 years that will provide maximal benefit for the scientific community and for key policy-adjacent communities
including HTAP and AMAP. HTAP3 Fires is aimed at providing fresh understanding of the atmospheric and environmental
impacts of fires and providing the foundation for sound policy decisions

## Data and Code Availability

Not applicable.

## Acknowledgements

We thank the following people for contributing their ideas for this study: Sabine Eckhardt, Twan van Noije, Marianne
Tronstad Lund, Ilia Ilyin, Havala Pye, Siti Aminah Anshah, Keren Mezuman, Nikos Daskalakis, Solène Turquety, Matthew
MacLeod, Rosa Wu, and Terry Keating.

## Funding Sources

PZ acknowledges support from DOE ASR DE-SC0021250 and NASA 80NSSC21K1344. The contributions from MM were
funded by the German Federal Ministry of Education and Research (Funding Nr.: 01LN2207A, IMPAC²T). EBM would like
to acknowledge: The European Research Council under the Horizon 2020 research and innovation programme through the
ERC Consolidator Grant FRAGMENT (grant agreement no. 773051), the AXA Research Fund through the AXA Chair on
Sand and Dust Storms at the Barcelona Supercomputing Center (BSC), and the BIOTA project PID2022-139362OB-I00
funded by MICIU/AEI/10.13039/501100011033 and by ERDF, EU.

## Author Contributions

CHW was coordinating lead on this paper, TB and JWK represented the TF HTAP steering committee, and *all* authors co-
wrote the paper.


## Competing Interests

At least one of the co-authors is a member of the editorial board of Geoscientific Model Development.

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

**Appendix**

**Table A.1:** Complementary fire-related research activities.

| Name | Objective | website | notes |
|---|---|---|---|
| Biomass Burning Uncertainty: Emissions, ReactioNs, and Dynamics (BBURNED) | To coordinate fire research community towards better understanding fire variability and uncertainty, particularly as it relates to atmospheric chemistry | https://www2.acom.ucar.edu/bburned | An International Global Atmospheric Chemistry (IGAC) activity |
| Arctic Monitoring and Assessment Programme (AMAP) | Inform the Arctic Council through science-based, policy relevant assessments regarding pollution and climate change issues | https://www.amap.no/about | Expert groups on SLCFs, POPs, Hg, Local v Long Range.<br><br>SLCF EG may use these HTAP experiments for a future AMAP report. |
| Arctic Black Carbon impacting on Climate and Air Pollution (ABC-iCAP) | Creation of fire management scenarios for Arctic Coucil countries/states | https://abc-icap.amap.no/ | |
| WMO Vegetation Fire Smoke Pollution Warning Advisory and Assessment System (VFSP-WAS) | To enhance the ability of countries to deliver timely and quality vegetation fire and smoke pollution forecasts, observations, information and knowledge to users through an international partnership of research and operational communities | https://community.wmo.int/en/activity-areas/gaw/science/modelling-applications/vfsp-was | |
| International Association of Wildland Fire (IAWF) | Organizes large-scale conferences around wildfire | https://www.iawfonline.org/ | |
| integrated Land Ecosystem-Atmosphere processes Study (iLEAPS) | Recently conducted a meeting on fires in south Asia, focusing on the prescribed fires and its modelling and planning next workshop in March. Carry out the conversion on the prescribed fires and its impact on air quality, health and modelling including fire emission estimate. | https://ileaps.org/future-earth and https://www.tropmet.res.in/204-event_details | |



| Arctic Community Resilience to Boreal Environmental change: Assessing Risks from fire and disease (ACRoBEAR) | To predict and understand health risks from wildfire air pollution and natural-focal disease at high latitudes, under rapid Arctic climate change, and resilience and adaptability of communities across the region to these risks. | www.acrobear.net | Integrating health data and knowledge, community knowledge and stakeholder dialogue, with satellite and in-situ observations, and numerical modelling. |
|---|---|---|---|
| Air Pollution in the Arctic: Climate Environment and Societies (PACES) | Review existing knowledge and foster new research on the sources and fate of Arctic air pollution, and its impacts on climate, health, and ecosystems | https://igacproject.org/activities/PACES | IGAC/IASC initiative. mproving knowledge of high latitude forcing from fire emissions. Key questions around ageing of fire plumes, mixing with anthropogenic pollution following export (e.g. POLARCAT cases). $NO_y$ speciation, BC ageing, how these vary between models... Need for improved observational constraint on these processes |
| AeroCom | | https://aerocom.met.no, and more specifically e.g. https://aerocom.met.no/node/110 and https://aerocom.met.no/node/115 | |
| The Fire Model Intercomparisons Project (FireMIP) | Systematic examination of global fire models, which have been linked to different vegetation models. Relevant for ESM/coupled fire simulations. | https://gmd.copernicus.org/articles/10/1175/2017/ | ISIMIP3 [the Intersectoral Impacts MIP phase 3] which FireMIP is now merging with - ISIMIP3b [simulations currently in progress] will produce multi-model projected future fire emissions for different SSP scenarios) |
| Support for National Air Pollution Control Strategies (SNAPCS) | Part of this project involves investigating the impact of local and long-range transport of fire-related pollutants on the UK. There is particular interest in implications for health/air quality, and model development | | Project involvement from the UKCEH, Imperial College London, EMRC and DEFRA |
| European Network on Extreme fiRe behaviOr (NERO) | bringing together wildfire researchers and practitioners to advance the current state of the science, thus making a crucial step in improving fire management, firefighter training and safety, and public safety planning Science-based wildfire management | https://www.cost.eu/actions/CA22164/ | European Cooperation in Science and technology (COST) Action |





| FLARE: Fire science Learning AcRoss the Earth system' (workshop) | The goal is to develop a roadmap for coordinated wildfire research for the next 5- 10 years. | https://futureearth.org/initiatives/funding-initiatives/esa-partnership/ | Held 18-21 September 2023. Article: https://futureearth.org/2023/12/13/reflections-from-the-fire-science-learning-across-the-earth-system-flare-workshop/ |
|---|---|---|---|
| AerChemMIP2 in CMIP7 | Historical and future climate change simulations focused on aerosols and trace gas chemistry for CMIP7. | | Will include a focus on wildland fires and biomass burning. Simulation design in 2024. |





**Table A.2** Model characteristics of potential participants.
PI = principal investigator(s), BC/IC=boundary/initial conditions. Model types (see Section 4.1 for acronyms).

| Organization | PI | Model(s) | Type | species | domain | Resolution (spatial and temporal) | BC/IC |
|---|---|---|---|---|---|---|---|
| University of Hertfordshire, Centre for Climate Change Research | Ranjeet S Sokhi | WRF/CMAQ, WRF-Chem | CTM | $PM_{2.5}$ (components), $O_3$, $NO_2$ | Europe, CORDEX | 5-10km over Europe, hourly, daily, monthly (for future projections) | CAM-chem, ECMWF, GFS |
| Environment and Climate Change Canada Climate Research Division | Cynthia Whaley, Knut von Salzen, David Plummer, Vivek Arora | CanAM-PAM, CMAM, CanESM CLASSIC (global & regional) | ESM and CCM | $PM_{2.5}$, $O_3$, $NO_x$, $CH_4$ | Global, Arctic, North America | T64, typically 3-hourly to monthly output | CanESM provides BC/IC for CanRCM |
| NSF National Center for Atmospheric Research Atmospheric Chemistry Observations and Modelling | Louisa Emmons, Rebecca Buchholz, Douglas Hamilton | MUSICAv0 (CESM/CAM-Chem) | | $O_3$, $NO_x$, $CO$, VOCs, $PM_{2.5}$ and speciated aerosols, metals | Global, U.S. | 12-km over US (and other regions), 1-deg global | |
| NASA Goddard/University of Maryland | Min Huang | WRF-Chem (NASA version, LIS) | ESM | $O_3$, PM, $CO$, $NO_x$, VOCs | Eastern US | 10 km or finer | CAMS, CAM-Chem/WACCM |
| NASA Goddard Institute for Space Studies | Keren Mezuman, Kostas Tsigaridis | NASA GISS ModelE | ESM | $CO_2$, $CO$, HCHO, $CH_4$, Acetone, Alkenes, Paraffin, $SO_2$, $NO_x$, $NH_3$, aerosols/$PM_{2.5}$ (organic, black carbon) | global | Cubed-sphere 1x1 effective resolution. 30 minutes to monthly | |
| Cyprus Institute Climate and Atmosphere Research Centre | Theo Christoudias | EMAC, WRF-Chem | CTM | PM (incl. Black carbon), $CO$, $O_3$ | Global, Middle East, North Africa | 20km over Middle East (2 mins), 1-deg global (15 mins) | GFS/WACCM |
| University of Bremen, Institute of Environmental Physics | Nikos Daskalakis, Sarah-Lena Meyer, Mihalis Vrekoussis | TM5-MP | | $PM_{2.5}$, $O_3$, $NO_x$, $NO_2$, $CO$, $CH_4$, VOCs, OA, speciated Aerosols | global | 1°x1°; 3-hourly to monthly | |
| UKCEH/Edinburgh University, | Damaris Tan, Stefan Reis | EMEP4UK, EMEP MSC- | CTM | $PM_{2.5}$ and components | Global, Europe, UK | 1km or 3km over UK, 27 km over | WRF with GFS/ERA5 |




| Institution | Author | Model | Type | Species | Region | Resolution | reanalysis |
|---|---|---|---|---|---|---|---|
| Atmospheric Chemistry and Effects/School of Chemistry | Mathew Heal, Massimo Vieno, Eiko Nemitz | W WRF | | | | Europe, 1 deg global resolution. Hourly output. | reanalysis |
| UK Met Office, Hadley Centre | Steven Turnock, Gerd Folbert, Joao Teixeira | UKESM1+ INFERNO | ESM | $PM_{2.5}$, $O_3$, $CH_4$ | Global | Global grid at 1.25 x 1.75 resolution (~140km) | |
| Lancaster University, Lancaster Environment Centre | Oliver Wild | FRSGC/UCI CTM | CTM | $O_3$, $NO_x$, CO, VOC, $CH_4$, gas-phase oxidants | global | Usually T42 (2.8x2.8 deg) but T106 (1.1x1.1 deg) feasible; output hourly/monthly | |
| Thailand Team (currently, King Mongkut's University of Technology Thonburi, Thammasart University) | Kasemsan Manomaiphiboon, Vanisa Surapipith | WRF-Chem | CTM | $PM_{2.5}$ (primary/secondary), $O_3$ | Upper Southeast Asia (with focus on Lower Mekong Region) | 4-12 km; output hourly | |
| IITM Pune, India, AQEWS Urban Air Modelling | Rupal Ambulkar, Sachin D, Gaurav Govardhan | WRF-Chem | CTM | $PM_{2.5}$, $PM_{10}$, CO | India | 10km over India run with daily output | GFS |
| CICERO Center for International Climate Research, Oslo, Norway | Marianne Tronstad Lund | OsloCTM3 | CTM | $O_3$, $NO_x$, CO, VOCs, $PM_{2.5}$ and speciated aerosols | global | 2.25x2.25 deg (possibly 1x1 deg depending on no. of simulations/scope), 60 vertical layers. Monthly output, with option of 3-hourly | |
| DLR Institute of Atmospheric Physics, Earth system modelling | Mariano Mertens | EMAC, MECO(n) | | $O_3$, $NO_x$, CO, VOCs, $PM_{2.5}$ and speciated aerosols | Global, Europe, West Africa | | |
| NOAA Geophysical Fluid Dynamics Laboratory (GFDL) | Meiyun Lin | GFDL AM4VR | | $O_3$, PAN, $NO_x$, CO, VOCs, $PM_{2.5}$ and speciated aerosols | Global, North America | ~13 km over North America, 25-50 km over Europe, and 50-100 km over Asia. Monthly, daily | |
| MIT, Earth, | Noelle Selin, | GEOS-Chem | CTM | PAHs, Hg | global | 2x2.5, 47 vertical | |





| Institution | Contact | Model type | Model | Species | Region | Resolution / output | Notes |
|---|---|---|---|---|---|---|---|
| Atmospheric and Planetary Sciences | Lexia Cicone, Eric Roy | | | | | layers, monthly outputs | |
| Meteorological Research Institute, Japan Met Agency (MRI-JMA), Dept of atmosphere, ocean, and Earth system modelling research | Naga Oshima | ESM | MRI-ESM2, TL159 | $PM_{2.5}$, speciated Aerosols, $O_3$, $NO_x$, $NO_2$, CO, $CH_4$, VOCs | global | AGCM: TL159 Aerosol: TL95 Ozone: T42 | |
| Institut National Polytechnique Felix Houphouet-Boigny (INP-HB) Department of Forestry and Environment | Jean-Luc Kouassi | | | $PM_{2.5}$, $O_3$, $NO_2$, CO, $CH_4$ | Global, West Africa | | |
| Space Research of the Netherlands, Earth Dept. | Helene Peiro, Ilse Aben, Ivar van der Velde | Inverse model | TM5-4DVar zoom | CO | Global, North America, Europe, Africa | Global 3x2, Regional 3x2 and 1x1; daily to monthly | |
| Norwegian Met Inst, Research Dept | Jan Eiof Jonson | | EMEP | $O_3$, $NO_x$, CO, NMVOCs, PM | Global, Europe | daily to annual output | |
| NILU, Atmospheric Chemistry | Sabine Eckhardt, Nikolaos Evangeliou | Lagrangian Transport Model | FLEXPART | CO, BC | global | 0.5 degrees, 3h resolution | |
| Finnish Meteorological Institute, Atmospheric Composition Research | Mikhail Sofiev, Rostislav Kouznetsov, Risto Hanninen, Andreas Uppstu, Evgeny Kadantsev | | IS4FIRES-SILAM | $O_3$, $NO_x$, CO, VOCs, $PM_{2.5}$ and speciated aerosols | Global to local | Several options, e.g., global 5-days forecast 10km for fire PM, 20km for full AQ tropo-strato, 10km Europe. Multi-annual reanalysis up to 50km global | Global nested, CAMS |
| Tsinghua University, School of Environment | Shuxiao Wang, Bin Zhao, Yicong He, Lyuyin Huang | ESM and CTM | CESM, WRF-Chem | $PM_{2.5}$, $O_3$, $NO_x$, $NO_2$, CO, $CH_4$, VOC, OA | Global, Southern China and SE Asia | 0.9x1.25 hourly, daily, monthly. 27km overall domain with 9km | |





| Institution | Author | Model | Type | Species | Domain | Resolution / Output | Driver |
|---|---|---|---|---|---|---|---|
| University of Augsburg, Faculty of Medicine | Christophe Knote, Bin Zhou | WRF-Chem | CTM | $PM_{2.5}$, OA | Europe | nested domain; hourly; 20km and 2km nest, hourly | GFS, CAM-Chem |
| Jožef Stefan Institute / MSC-E, Dept of Environmental Sciences | Oleg Travnikov | GLEMGLEOS | Multi-media POPs (?) | POPs, Hg, metals | Global, Europe | 1x1 degrees, monthly or daily output | |
| Peking University, Lanzhou University | Jianmin Ma, Tao Huang | CanMETOP, CMAQ | Long-range atmospheric physical transport and CTM | POPs, heavy metals | Global, China, North America | from 10 km to 1°x1°. Hourly, daily, and yearly | |
| Sorbonne Université, LATMOS/IPSL | Solène Turquety | CHIMERE | CTM | $O_3$, CO, VOCs, PAN, $NH_3$, aerosols | Northern Hemisphere, Europe | 1°x1° hemispheric, 10km Europe | CAMS |
| Stockholm University | Matthew MacLeod | BETR Global | Multi-media POPs | POPs, PAHs | Global | 3.75° x 3.75°, weekly or monthly output | |

Table A.2 continued





| Organization | meteorology | Anthro emissions | Fire emissions | Natural emissions | Simulation length | Fire plume height | References |
|---|---|---|---|---|---|---|---|
| University of Hertfordshire, Centre for Climate Change Research | | CAMS regional | CAMS GFAS, NCAR FINN | MEGAN | | | |
| Environment and Climate Change Canada Climate Research Division | Free-running or nudged to ERA-Interim | CMIP6, ECLIPSEv6b | CMIP6/GFED4 or use online interactive fire module from CLASSIC | | Season to multi-decade | Climatological distribution based on AEROCOM, or online with CFFEPS plume height scheme | |
| NSF National Center for Atmospheric Research Atmospheric Chemistry Observations and Modelling | Nudged to MERRA2 reanalysis | CAMS | FINN, QFED, or other | MEGAN online in CLM, prognostic sea spray and dust | Season to multi-year | | wiki.ucar.edu/display/camchem/Home |
| NASA Goddard/University of Maryland | WRF, initial/boundary conditions from NARR | CAMS (multi-year), HTAPv3 for 2018 | QFED, plume rise | Online biogenic and lightning | case studies and multi-year warm seasons | | Huang et al. (2022; 2024) |
| NASA Goddard Institute for Space Studies | | CMIP6 | GFED4s, pyrE (interactive fire model) | MEGAN, wind driven sea salt and dust, lightning | | | |
| Cyprus Institute Climate and Atmosphere Research Centre | | CAMS, EDGAR | FINN or GFED or other | MEGAN, lightning, dust/sea salt | 2010-present | | |
| University of Bremen, Institute of Environmental Physics | | CMIP6 | GFEDv3,GFEDv4,CMIP6 | MEGAN-MACC | Multi-year | | |
| UKCEH/Edinburgh University, Atmospheric | WRF, with reanalysis from GFS/ERA5 | NAEI for UK, CEIP for Europe, HTAP | NCAR FINN (v1.5 and transitioning to v2.5) | Online BVOCs, soil NOx, volcano, seasalt | Month to multi-year | Emissions evenly distributed | Simpson et al. (2012); Vieno et al. |





| Institution / Group | Meteorology | Anthropogenic emissions | Fire / biomass emissions | Natural / biogenic emissions | Timescale | Vertical distribution | Reference |
|---|---|---|---|---|---|---|---|
| Chemistry and Effects/School of Chemistry | depending on version/setup. Nudged every 6 hrs, 1 deg res. | (2010) for global | | dust | | over 8 lowest vertical layers (Simpson 2012) | (2016) |
| UK Met Office, Hadley Centre | | Mainly CMIP6, but flexible | can run with prescribed emission datasets or use online interactive fire model INFERNO | various and depends on the configuration setup but mostly interactive for dust, BVOCs, sea salt, DMS | Typically years, multi-year, multi-decade | | |
| Lancaster University, Lancaster Environment Centre | Driven by ECMWF-IFS cy38 met at TL159L60 3-hr resolution | flexible | flexible | MEGAN | Single-year/multi-year | Surface/PBL emissions only | Wild (2007) |
| Thailand Team, KMUTT | | HTAP/CAMS, locally adjusted | FINN and others | MEGAN | Sub-seasons, selected events | | |
| IITM Pune, India, AQEWS Urban Air Modelling | | EDGAR-HTAP | FINNv1.5 | MEGAN | Air quality forecast for 10 days | | |
| CICERO Center for International Climate Research, Oslo, Norway | | Flexible, but currently CEDS and ECLIPSE most used. | GFED4 | MEGAN (online or offline) | Single-year/multi-year, likely time slice for selected years on longer timescales. | | |
| DLR Institute of Atmospheric Physics, Earth system modelling | Free running or nudged (ERA5) | flexible | flexible | Lightning NOx, air-sea exchange, dust, biogenic and soil-NOx | | | Jöckel et al. (2016; 2010) |
| NOAA Geophysical Fluid Dynamics Laboratory (GFDL) | Driven by observed SSTs or nudged to reanalysis winds | CEDS-v2021-04-21 | GFED4 (daily or monthly), but flexible | Interactive MEGAN BVOCs; Interactive dust coupled to vegetation cover; Lightning NO coupled to | Multi-year, multi-decade | Distributed vertically up to 6 km, based on an injection height climatology from MISR | Lin et al. (2024a,b) |





| Institution | Meteorology | Anthropogenic emissions | Fire emissions | subgrid convection | Years | Notes | Reference |
|---|---|---|---|---|---|---|---|
| MIT, Earth, Atmospheric and Planetary Sciences | | PKU | PKU | PKU | | | |
| Meteorological Research Institute, Japan Met Agency (MRI-JMA), Dept of atmosphere, ocean, and Earth system modelling research | Free-running or nudged to JRA55 | CMIP6, ECLIPSEv6b | CMIP6/GFED, GFED | | | | |
| Institut National Polytechnique Felix Houphouet-Boigny (INP-HB) Department of Forestry and Environment | ERA5 | CMIP6 | GFED4 | | | | |
| Space Research of the Netherlands, Earth Dept. | mainly ECMWF | CAMS | GFED4.1s, or GFED5, or GFAS | | | IS4FIRES | |
| Norwegian Met Inst, Research Dept | ECMWF | Variable, mainly EMEP | FINN, GFAS | | | | |
| NILU, Atmospheric Chemistry | ECMWF, ERA5, CESM GCM, WRF, … | ECLIPSEv6b | GFED, GFAS | | Years | From GFAS emissions dataset | Pisso et al. (2019) |
| Finnish Meteorological Institute, Atmospheric Composition Research | nudged to MERRA2 reanalysis data | CAMS, CMIP, ECLISE, EDGAR | IS4FIRES | MEGAN or own model (Europe) | From 5 day forecasts up to multi-decades in climate mode | Sofiev et al. (2012) | |
| Tsinghua University, School of Environment | NCEP FNL reanalysis data | Huang et al. (2023) for full-volatility organic, CMIP6 for other pollutants. Chang et al | Full-volatility organic emissions based on the burning area of GFEDv4; Other pollutants from GFEDv4. FINN: full-volatility organic emissions based | dust/biogenic/sea salt: calculated online / dust/biogenic/sea salt: calculated online | Multi-year (2015-2020), 2018 | Daily fire info from FRP and met data. Freitas plumerise scheme | |





| Institution | Meteorology | Anthropogenic emissions | Fire emissions | Other emissions | Timescale | Notes | References |
|---|---|---|---|---|---|---|---|
| | | (2022) for full-volatility organic ABaCAS-EI for other pollutants | on the burning area of GFEDv4 | | | | |
| University of Augsburg, Faculty of Medicine | nudged to GFS during spinup above PBL | EDGARv5 + national German inventory | FINN | online | Season to multi-year | | |
| Jožef Stefan Institute / MSC-E, Dept of Environmental Sciences | ECMWF, WRF | EMEP, EDGAR, PKU | FINN, MCHgMAP, PKU | MCHgMAP PKU | Multi-year | | |
| Peking University, Lanzhou University | ECMWF, NCEP FNL | PKU, LZU, EDGAR | PKU, LZU | | Month to multi-year | Gaussian plume model to distribute to 3km height | Luo et al. (2020); Song et al. (2022) |
| Sorbonne Université, LATMOS/IPSL | ERA5, WRF | CAMS | CAMS GFAS, APIFLAME | MEGAN, dust, sea salt, lightning | Seasonal | Satellite observations, GFAS plume height | Menut et al. (2021); Turquety et al. (2020) |
| Stockholm University | Driven by ECHAM 5 model outputs | Flexible | Flexible | Flexible | Multi-year, multi-decade | Emission to boundary layer or free troposphere | MacLeod et al. (2011) |