# Peer review of "HTAP3 Fires: Towards a multi-model, multi-pollutant study of fire 1"

_Geoscientific Model Development, 2024_

## Author Response (AR2)

**HTAP Fires: Response to reviewers**

- Reviewer comments are in *italic*
- Authors' responses start with [AR]

Review #1:

*This paper presents a proposed protocol for a new set of HTAP experiments, focused on biomass burning. This paper is overall well written and provides a comprehensive description of the protocol. I have however a number of points and questions that would warrant being clarified.*

[AR]: Thank you for your thorough review.

*Line 118: What aspect of uncertainty is being tested here? Most likely just structural uncertainty since initial conditions are not really considered in the protocol*

[AR]: Initial condition uncertainty is not a major concern as our assessment of fire impacts is typically not limited by it (unlike weather forecasting). We focus here on process/parametric uncertainty (what is emitted? how high is it lifted? what chemistry does it undergo?) as well as structural uncertainty (how are processes implemented in models?). Section 4.5.3 is associated with this question and explains and expands on this aspect of the uncertainty.

*Line 120: Isn't there a risk that this pushes towards a parameter optimization? Isn't it more important to be right for the right reasons?*

[AR]: While we understand this concern, we believe that the associated sensitivity experiments described in Sections 4.5.3 and 5.3.1 will help identify which processes or parameterizations are needed to match observations well, and without which we can't match reality adequately. There is a range in complexity of fire process parameterizations in models, and in some cases, these experiments may reveal that a particular parameterization represents reality better than other parameterizations (e.g. one that represents the physics of plume rise or the chemistry in the plume better). In the revised manuscript, we include a "and why?" to this question to emphasise that we aim to analyse these results thoughtfully.

*Line 210: Critical in what aspect?*

[AR] We have rephrased this sentence to clarify: "BC accounts for about 10% of smoke plume mass and is the **largest** contributor to aerosol radiative forcing (RF) (Veira et al., 2016)"

*Line 260: I am not sure "congener" is the right term here*

[AR] We have removed that word, and rephrased as: "... sources of benzo[a]pyrene (BaP), a PAH with high carcinogenicity, accounting for …"

*Line 284: That is true that isotopic measurements might shed some light in this question of fingerprinting. However, on the modeling side, it seems to me that there might be ways of*

*using tagging methods to isolate specific contributions. It would not work with all tracers, but should work with some, especially within a few days of transport/chemistry from emissions.*

[AR]: Thank you for noting that models can use tagging methods to identify sources, which could then be compared to sources quantified by isotopes in the measurements. Indeed, our proposed emissions perturbation experiments (Section 4.5.2) are another method for source attribution and the one we decided on for this project. This sentence about isotopic measurements for metals in aerosols was mainly to provide the context that there is a gap in using this measurement-based method for the fire source, in particular.

*Line 334 and Table 1: RF is usually estimated as the change in radiative fluxes with respect to a reference period/composition. What is the reference here? No biomass burning? Are those Instantaneous radiative forcing calculations?*

[AR]: The reference period/composition is different for each study cited in this paragraph and Table 1. Also the methods (instantaneous/stratospherically adjusted flux change) differ. We made this more clear in this section of the revised manuscript.

*Lines 482-484: the sentence does not quite make sense*

[AR]: We have rephrased this sentence to: "Conversely, the minor sources that regional and national inventories had that were not present in EDGAR6.1 (e.g. …) were included in the HTAP v3 mosaic."

*Lines 492-497: The use of those scenarios continues to make a comparison with CMIP difficult. And it is noted later in this paper the importance of connection with AerChemMIP2. What is the justification for using those scenarios?*

[AR]: While it's true these IIASA GAINS scenarios are difficult to compare with CMIP scenarios, they were used in the recent AMAP SLCF assessment report (AMAP, 2021, where they are called ECLIPSEv6b), and also shown to be much more realistic in the near-term than CMIP6 emissions in (Ikeda et al, 2022), as they were based on more up-to-date emissions changes in Asia. The IIASA GAINS scenarios are also more focused on air quality policies than CMIP scenarios, and will be used in the HTAP3 OPNS modelling project that we'd like to be consistent with, maximizing the policy relevance for CLRTAP. We have added this justification to Section 4.2.1 in the revised manuscript.

References:

AMAP Assessment 2021: Impacts of Short-lived Climate Forcers on Arctic Climate, Air Quality, and Human Health, Arctic Monitoring and Assessment Programme (AMAP), Tromsø, Norway, viiiC 324 pp., https://www.amap.no/documents/doc/amap--assessment-2021-impacts-of-short-lived-climate-forcerson-arctic-climate-air-quality-and-human-health/3614, 2021.

Ikeda, K., Tanimoto, H., Kanaya, Y., Taketani, F.: Evaluation of anthropogenic emissions of black carbon from East Asia in six inventories: constraints from model simulations and surface observations on Fukue Island, Japan, Environ. Sci. Atmos., 2, 416-427, 2022, doi:10.1039/D1EA00051A

*Lines 500-504: what is the rationale for not harmonizing? This seems like a serious oversight that makes the comparison of the two simulations much less meaningful*

[AR]: The rationale for not harmonizing the historical & future anthropogenic emissions is that the HTAPv3.1 historical emissions and the GAINS future emissions are from very different sources. The HTAPv3.1 historical emissions are based on nationally reported emissions, while the GAINS future emissions come from the GAINS model, which includes its own data sources that only partially overlap with the information used by countries to report their own emissions. Any simple harmonisation of these datasets would either lead to historical emissions which are not consistent with the nationally reported totals, or future emissions which are not consistent with the policy measures represented in the GAINS model. In both cases, this would reduce the policy relevance of the datasets. A thorough and careful harmonisation of these datasets would require detailed consultations between the GAINS modellers and the national inventory builders for all of the regions contributing to HTAPv3.1, which is well beyond the scope of this exercise.

However, we understand your concern, so we have also clarified in the revised manuscript, that the historical & future biomass burning (BB) emissions *are* harmonized with each other, which ensures that the absolute (if not relative) differences for BB across the full time period (early 2000s to 2050) *are* meaningful. A new Section 5.5.4 has been added to clarify how to assess how fires and their impacts will change in the future, and we have added more detail there in the revised manuscript to ensure the future minus "present" analysis will be meaningful by comparing the 2015-2020 with 2045-2050 to each other with a consistent set of anthropogenic input emissions.

*Line 565: In that section, should there be a discussion of the potential emission of micro-plastics?*

[AR]: While microplastics can act as a vector for POPs, microplastics themselves are not a pollutant covered under HTAP, and while we aim to include "non-traditional pollutants" in this study, the scope of this project is already sufficiently large with the pollutants that we do include.

*Section 4.3.2.: this is just a description of CMIP6. How does that apply to the protocol discussed here?*

[AR]: Section 4.3 discusses the possible meteorological inputs for models, and this section summarizes potential sources of meteorology available for the future simulations (2015-2050). We have now revised this description to include more context for the HTAP Fires project in Section 4.3.2 and Section 5.2 in the revised manuscript.

*Figure 3: it seems to me that a region prone to very large fires such as the Mediterranean basin should have a box by itself. What is the rationale for such generic areas (other than the fact that's the way it's done for other HTAP projects)?*

[AR]: As mentioned on line 715-716 in the original manuscript, "the number of perturbation experiments can increase rapidly, so care is needed to prioritize and not define regions and sectors too finely" – this is the main reason. The BB community discussed the pros and cons of fine vs coarse regional definitions, and one additional justification for coarse regions was

that the seasonality and fire characteristics of the smaller regions help to identify themselves on their own, even in a coarsely defined box. For example, the western US has a different BB seasonality than the eastern US, and thus, they will be distinguishable in the results, despite the US being one large box combined with Mexico.

*Line 747: how long are the simulations intended to be?*

Those simulations are meant to be over 2015-2019 or 2003-2020 time periods, depending on models' computational expense. This information was given in Table 5 of Section 5.3 (Section 5 being the recommended plan), where we included this information:

| | | | |
|---|---|---|---|
| 5. Regional and sectoral emissions perturbations | Turn off all BB emissions for all species everywhere.

Turn BB emissions off in each region of Figure 3d, and each of the 2 sectors: agricultural burning and wildland fires, over the historical time periods in Table 3. | To quantify regional source/receptor relationships and uncertainties | High

Priority: Both fire sectors together; separate sectors if resources permit. |

…referring to Table 3:

**Table 3:** Simulation time periods, with options for different types of models.

| | Short option | Medium option | Long option |
|---|---|---|---|
| **Historical** | See Case studies (Sec 4.5.1) | 2015-2019 | 2003-2020 |

*Figure 4:  the color scheme makes it hard to identify the SAOM areas.*

[AR]: the SAOM (orange) areas are pretty small, and appear in northern Europe next to EFOM (purple). However, in the revised manuscript, we replace this figure with a new one with a different colour scheme.

*Line 758:  this is an awkward sentence.*

[AR]: Thanks for highlighting this. In the revision we have re-written this part to make it clearer and more straightforward. It now reads as: "Fires are highly variable, depending on surface and atmospheric conditions (e.g. temperature, winds, soil moisture), vegetation type, and the evolution of the fire over time (e.g.changes in size and intensity). This variability makes fire processes and their representation in models an ongoing challenge for researchers. The uncertainties in simulating fire processes from emissions (magnitude, timing, and vertical distribution) to plume chemistry and transport, to deposition, all contribute to the uncertainty in simulating fire impacts"

*Line 821: the only thing that these experiments will do is to show how sensitive the results are to deposition.  Will there be an attempt to use deposition data where available?*

[AR]: Yes, in Section 4.4 we mentioned that "Cross-disciplinary satellite (atmospheric, land cover, water quality, etc) and in-situ data can be used to evaluate modelled deposition results, helping identify weakness in individual models and reduce uncertainty in impact assessments (e.g., Fu et al., 2022; Huang et al., 2024)", and Section 5.5.1 includes discussion of model evaluation with observations.

*Lines 843-845: the use of 4 years is better than a single one, but there is then the covariance of fire and meteorology. One could tease out the separate roles by keeping one constant, in addition of having them both vary.*

[AR]: Yes, we will do this in Exp 6 (described on lines 919-920 of the original manuscript).

*Line 931: how are the "radiative forcing output for climate impacts" computed? Are those from a double call to the radiation?*

[AR]: Yes, radiative forcing is computed in some models by a double-call to the radiation scheme, in which one call excludes radiatively active atmospheric components of interest, and the other includes full atmospheric radiative effects. The difference in radiative fluxes can then be used to diagnose a forcing. For models where this is not possible or routine, an offline radiation scheme can be used to compute forcing from gridded model output of trace gases or aerosol from two model scenarios (e.g. Hollaway et al., 2017). For some species (e.g. ozone), pre-computed radiative kernels (Rap et al., 2015) can be used offline to directly translate changes in atmospheric composition fields into estimates of radiative effect / forcing (e.g. Rowlinson et al., 2020). We add some of these details to the revised manuscript, section 5.5.3.

References:
Hollaway, M. J., S. R. Arnold, W. J. Collins, G. Folberth, and A. Rap (2017), Sensitivity of mid-nineteenth century tropospheric ozone to atmospheric chemistry-vegetation interactions, J. Geophys. Res. Atmos., 122, 2452–2473, doi:10.1002/2016JD025462.
Rap, A., N. A. D. Richards, P. M. Forster, S. A. Monks, S. R. Arnold, and M. P. Chipperfield (2015), Satellite constraint on the tropospheric ozone radiative effect, Geophys. Res. Lett., 42, 5074–5081, doi:10.1002/2015GL064037.
Rowlinson, M. J., Rap, A., Hamilton, D. S., Pope, R. J., Hantson, S., Arnold, S. R., Kaplan, J. O., Arneth, A., Chipperfield, M. P., Forster, P. M., and Nieradzik, L.: Tropospheric ozone radiative forcing uncertainty due to pre-industrial fire and biogenic emissions, Atmos. Chem. Phys., 20, 10937–10951, https://doi.org/10.5194/acp-20-10937-2020, 2020.

*Section 5.5.2: will that take into account potential changes in population size and age distribution for the future simulations? Some analysis have shown that this was the largest factor (e.g. https://www.nature.com/articles/s41893-022-00976-8)*

[AR]: Thank you for the reference, which we have added to the revised manuscript. Health impacts of long-term (chronic) exposure to ambient ozone and PM2.5 will be estimated using standard health impact assessment modelling frameworks, based on data from epidemiological cohort studies. These approaches estimate the increased relative risk of health impacts caused by chronic ambient pollutant exposure above a counterfactual level for different age groups. These methods consider population count and distributions, and population age structure and baseline mortality rates. Current and future cause-specific baseline mortality rates and population age structure are available from International Futures (IFs) (Frederick S. Pardee Center for International Futures, 2021), and global gridded population count for present and future scenarios consistent with SSP scenarios are available from Jones and O'Neill (2016, 2020). We have previously used these datasets to estimate future PM2.5-attributable mortality in SSP scenarios (Reddington et al., 2023).

Models that specifically consider the chronic health impact of wildfire smoke are currently not available in the literature, however there are some recent studies that have accounted specifically for increased health risks resulting from wildfire emission exposure (e.g. Chen et al., 2021). We will update our health impact assessment framework using data from such studies if the underlying models and data become openly available.

The main focus of the current modelling project will be to provide the required atmospheric pollutant concentrations to the health analysis experts.

Review #2:

**General Comments**

*The authors describe the motivation, scope, and experimental design of a planned multi-model, multi-scale, and multi-pollutant study that is aimed at improving our understanding of wildland fires and their impacts on air quality. The planned work under HTAP3 Fires is indeed a very worthwhile endeavor, and sections 2 and 3 of the manuscript provide an excellent summary of the research needs and how the expected results of this effort can help address important science and policy questions. While sections 4 and 5 are intended to provide details on how the planned activity will be conducted, reading these sections created the impression that some aspects of the study are still fluid and may be decided at a later time. Several instances of such ambiguities between firm plans vs. potential avenues for research are noted in my detailed comments below and should be clarified in the revision. While I appreciate the benefit of publishing an experimental design paper now to anchor the activity, the drawback of this early publication is that a fair amount of information which would ordinarily belong in an experimental design paper (e.g. a definitive list of selected – rather than potential - case studies, protocols for coordinating modeling and analyses across global and regional scales, infrastructure for model evaluation) are not yet finalized, thereby limiting the utility of this paper as a definitive reference for the planned work. This doesn't argue against publishing the manuscript at this stage, but should be acknowledged in the discussion. In general, I find that the expansive scope of the HTAP3 Fires modeling activity, as outlined in this paper, is a great strength but could also pose challenges when it comes to analyzing, synthesizing, and interpreting the results. I am certainly looking forward to these results and wish the organizers and participants of the activity best of success.*

[AR]: Thank you for your thorough review and good wishes! While Section 4 may appear to include some ambiguity, this is because its principal purpose is to document the options that were considered in the experiment design process. Section 5 presents the definitive plan, after the range of options presented in Section 4 were thoroughly considered. In this way, we aim to provide justification of the final decisions on experiment design that are presented in Section 5. We have now revised the end of the introduction and the conclusions to make sure that this is clear.

Also, in the time since the initial submission of this paper (~6 months ago), several details have been firmed up or clarified, so our revised paper will include more definitive details (e.g. in Sections 5.2 and 5.3), improving the utility of this paper. Indeed, the experiments are nearly ready to be launched at this time.

Finally, one additional important note that applies to a few of the reviewer comments below is that an additional technical guidance document will be circulated in the near future to modelling centres who will participate in HTAP Fires. It will contain additional firm technical notes (such as a breakdown of the sub-experiments and their priorities, naming convention

for model output files, where and how to upload model output, etc) that, while important for modellers, are not necessary to be documented in this peer-reviewed paper.

***Specific Comments***

*Line 67: "analyses" instead of "analysis"?*

[AR]: Thank you, we have corrected this.

*Line 73: remove the comma after the semicolon*

[AR]: Thank you - done!

*Line 128: please clarify what is meant by "differ based on initial pollutant focus"*

[AR]: We meant that some models, like CTMs and air quality-focused models have a focus on NOx, VOCs, O3, CO, and PM2.5, whereas other models, like ESMs have a focus on GHGs and PM, and yet other model types have a focus on toxics like Hg, PAHs, POPs, etc. Their *focus* is different, but they may include common pollutants, which means we can cross-compare their results. In the revised manuscript, we have re-written this bullet to be more clear.

*Lines 160 – 162: This reads like a repeat of the discussion for chemistry in freshly emitted plumes on lines 152 – 153*

[AR]: Thank you for pointing out this repetition. The revised manuscript now merges these two sentences into one.

*Line 227: remove ", which" between "(POPs)" and "are synthetic pollutants"*

[AR]: Removed.

*Lines 294 – 295: Does the Xu et al. (2023) study also refer to worldwide numbers, like the Johnston et al. (2012) study referenced in the previous sentence?*

[AR]: Yes, they do. From Xu et al. (2023): "During the period 2010–2019, 2.18 billion people were exposed to at least 1 day of substantial LFS air pollution per year, with each person in the world having, on average, 9.9 days of exposure per year". We clarified this in the text: "Xu et al (2023) estimated each person in the world having an average of 9.9 days of smoke exposure from 2010-2019"

*Lines 321 – 322: This sentence needs rewording, starting with "'is highly uncertain" which is a phrase used at both the beginning and end of this portion of the sentence.*

[AR]: Thanks for pointing this out. We have revised the sentence to be clearer: "As radiative forcing is typically expressed as a change relative to the preindustrial era, and the magnitude of preindustrial fires is highly uncertain, there is a factor of 4 uncertainty in RFs from fires (Hamilton et al, 2018; Wan et al, 2021; Mahowald et al. 2023)."

*Lines 334 – 326: Please revisit the structure of these two sentences: "Though, most studies focus on … only. However, …". Maybe remove "though" at the beginning of the first sentence?*

[AR]: Thank you for pointing out these two incomplete/awkward sentences. We have revised them: "Though, most studies focus on specific components or regions for wildfire RF (e.g., Mao et al., 2012; Chang et al., 2021, Mubarak et al., 2023), Ward et al. (2012) conducted a comprehensive global analysis of wildfire emission's RF, encompassing all components."

*Line 340: Insert "source of" before cloud condensation nuclei?*

[AR]: Thank you. This has been corrected.

*Line 349: Please define rBC*

[AR]: "refractive black carbon (rBC)" has been added to the revised manuscript.

*Lines 349 – 350: It is unclear what the last part of the sentence "might help to reduce these discrepancies between the models" refers to – new measurement data? If so, consider breaking this into a new sentence, after "Dobracki et al., 2023)". "In addition, new data might also help to …"*

[AR]: Thank you for bringing this to our attention; yes, it refers to new measurement data. This sentence has been split into two sentences in the revised manuscript, with the second one beginning with "New data on long-range transported aerosol …"

*Lines 356 – 357: "Note that for some regions, actions have been taken to reduce such impacts which may or may not be accounted for or represented well in models." – which types of models does this refer to? As written, it's a rather vague statement.*

[AR]: We meant that some human intervention/fire management practices are different regionally and change over time, and models often do not include those nuances, or include them in a coarse or simplified manner. We have revised the sentence to be clearer: "Note that human intervention/fire management practices to reduce these fire impacts vary by region, but those activities may or may not be accounted for or represented well in atmospheric and Earth system models"

*Line 421: Suggest not referring to HCHO as a by-product of SOA, given that it is also emitted as well as produced as part of the atmospheric oxidation of isoprene and monoterpenes.*

[AR]: Thank you for pointing this out. We removed that part of the sentence in the revised manuscript.

*Lines 445 – 447: Suggest also mentioning chemistry transport models that ingest meteorology from prognostic meteorological models employing nudging– the current description would seem to cover models like WRF/Chem or GEM-MACH with the first example, and models like GEOS-Chem with the second example, but not modeling systems like WRF-Chimere where WRF simulations (employing nudging towards reanalysis fields) are used as input to Chimere.*

[AR]: We thank the reviewer for pointing this out. In hindcast or historical simulations, regional prognostic meteorological models can ingest (or downscale) reanalysis data in two different ways, i.e., with or without nudging. The former deals only with initial and boundary conditions. The latter dynamically nudges model output towards selected reanalysis fields, which helps preserve or maintain the underlying meteorological conditions generally at meso- and synoptic scales. Modelers in the HTAP3-Fires can weigh which way is more justifiable to their purposes. However, nudging in online coupled modelling may not be encouraged for some applications since it potentially obscures or affects interactions between meteorology and chemistry. We have revised the sentences to acknowledge these as follows: "Of note, in hindcast or historical simulations, regional prognostic meteorological models can ingest (or downscale) reanalysis data in two different ways, i.e., with or without nudging. The former deals only with initial and boundary conditions. The latter dynamically nudges model output towards selected reanalysis fields, which helps preserve or maintain the underlying meteorological conditions generally at meso- and synoptic scales. Modelers in the HTAP3-Fires can weigh over which way is more justifiable to their purposes. However, nudging in online coupled modelling may not be encouraged for some applications since it potentially obscures or affects interactions between meteorology and chemistry."

*Line 480: Based on my reading of Section 2.3.2 of Crippa et al., it is not correct to refer to the U.S. portion of HTAPv3 as "official national inventories". Instead, these emissions were based on the EPA's Air QUAlity TimE Series Project (EQUATES), as described in Foley et al. (2023) (https://doi.org/10.1016/j.dib.2023.109022). As discussed in Crippa et al., "For each sector, a consistent methodology was used to estimate emissions for each year in the 16-year period, in contrast to the evolving methodologies applied in the triennial US National Emissions Inventories (NEIs) produced over that span."*

[AR]: Thank you. The US emissions in HTAPv3.1 are not exactly the same as the US NEI, but rather from a consistent time series produced by the US-EPA, who also produce the NEI. We have modified the text to read: "emission data from the respective national authorities" instead of "official national inventories".

*Line 485: "By September 2024, HTAP v3.1 …" To my knowledge, these emissions were not released in September 2024. Please update the timeline.*

[AR]: That is correct. We have revised the text to say January 2025, and we have also added a data availability section to the revised manuscript, as well as updated Table 4 in Section 5.2. Hopefully this also helps address your initial concern about the definitiveness of this model design paper.

*Line 491: "will be available from July 2024". To my knowledge, these emissions were not released in July 2024. Please update the timeline.*

[AR]: That is correct. We have revised the text to say January 2025 (for the future anthro emissions from GAINS LRTAP), and, similar to the preceding comment, we have also added a data availability section to the revised manuscript, as well as updated Table 4 in Section 5.2. Hopefully this also helps address your initial concern about this model design paper.

*Line 505: is there no plan to apply CTMs with historic meteorology for future emission scenarios?*

[AR]: There is a plan to use constant present day meteorology with changing fire emissions (e.g. lines 919–920 in the original manuscript), as well as a plan for CTMs to do future emission scenarios. This part of the text has been revised based on a comment by reviewer #1, and no longer appears in the revised manuscript.

*Lines 506 – 510: This section also needs to discuss how individual models will speciate VOC and PM2.5 emissions into their gas phase chemistry and aerosol mechanisms. Will reference speciation profiles be provided, or will this important decision be up to each group? How do the HTAPv3 (and GFAS) emissions handle intermediate-volatility organic compounds (e.g. https://acp.copernicus.org/articles/19/8141/2019/, https://doi.org/10.1016/j.oneear.2022.03.015, https://acp.copernicus.org/articles/23/13469/2023/, and references therein)? If these are not fully accounted for in the emission inventories, can groups account for such missing mass in their simulations by applying correction factors?*

[AR]: Thank you for pointing out this gap in information. We have revised Section 4.2 to mention that HTAPv3.1 provides speciation for PM and NMVOC, but not IVOCs. Modelling centres will use their own approach for IVOCs. GFAS4HTAP uses the speciation from NEIVA (https://doi.org/10.5194/gmd-17-7679-2024), and other speciation will need to be adapted for individual models.

*Lines 512 – 513: Please provide a reference for each of these inventories.*

[AR]: References have been added for each of these emissions datasets in the revised manuscript.

*Lines 517 – 518: Does this statement refer to the workshop discussions, the studies listed on lines 515 – 516, or both? Will FIRECAM be used in this study? If not, what is the motivation for mentioning it here?*

[AR]: The statement about FIRECAM being useful for comparison was discussed at the workshop and included in one of the intercomparison presentations/references (Liu et al, 2020). The motivation for mentioning it here was to emphasize that the process in selecting one of many BB datasets was done thoughtfully and robustly.

*Line 519, Table 2: This table should include all of the latest available major fire emission datasets listed at the beginning of Section 4.2.2. In addition, FEERv1.0-G1.2 is shown in the table but not named as one of the major fire emission datasets listed at the beginning of Section 4.2.2.*

[AR]: Thank you for this suggestion. In the revised manuscript, we make the text and Table 2 consistent with each other. Note that the revised table has been transposed to make room for more emission datasets.

*Lines 538 – 539: Why is GFAS (v1.4) mentioned here – are there plans for using it in later stages of HTAP3 Fires? If not, what is the relevance of mentioning it?*

[AR]: Both GFEDv5 and GFASv1.4 are mentioned in this section as existing, cutting edge, or soon-to-be-released emissions datasets that contain peat fire emissions. This section

documents all of the discussions and considerations that went into the final section of BB emissions since it's such an important parameter of this model design.

*Line 544 as well as Table 2: Please define FRP upon first usage. Right now, it is not defined until about line 750.*

[AR]: Thank you for pointing out this oversight. We now define this acronym on first use.

*Lines 546 - 548: It's not clear how the discussion of diurnal FRP information in GFASv1.4 is relevant to this project.*

[AR]: Our apologies. Later in this section, we have a heading for the Timing of Emissions, which explains the importance of the diurnal cycle of fire emissions. However, it comes up under the peat heading as a type of fire emissions, where the diurnal cycle is less apparent. We have the revised the manuscript as followed to put this into context:
"... and availability of information on the diurnal cycle (more on timing of emissions below). Unlike other types of wildland fire emissions, tropical and mid-latitude peat fires generally have a flat diurnal cycle…"

*Line 616: "but could be corrected to GFASv1.2" – is this a firm plan, or just a possibility? If it's only a possibility, what is the timeline and decision making process for determining whether this will done?*

[AR]: This is a case where it was a plan to be done at the time we submitted the original manuscript (~6 months ago), and now it's underway (we expect within weeks). We have revised the manuscript (Sec 4.2.3) to be less ambiguous about this.

*Lines 632 – 634: please double check the sentence structure following "across a wide range of scales". Usually "from turbulent mixing …" would be followed by "to …", but this is not the case here.*

[AR]: Thank you. We have revised to add the "to" to this sentence.

*Lines 634 – 636: please see my earlier comment on offline CTMs, which do not generate their own meteorology, but are not (directly) driven by reanalysis products, either, instead using meteorological fields from models like WRF that employ nudging.*

[AR]: This sentence has been revised to make it clearer: "Observation-based reanalysis datasets provide an important source of meteorological information needed to drive some of the models (included in Table A.2), but differences between these products, and between reanalyses and model-generated meteorology, provide an additional source of uncertainty."

*Lines 638 – 641: Will the choice be completely up to individual modeling groups?*

[AR]: Yes, in this section (4.3.1), we just discuss the options, and in Section 5.2 we encourage the use of ERA-5 for offline CTMs, but say that modellers may use their "preferred meteorology for historical simulations and ensure that they document this clearly".

*Lines 666 – 667: Satellite data products should not be referred to as direct observations, given that assumptions are invoked when generating such products. Please also discuss*

*how such products will be used in planned evaluation studies and to which extent such analyses will be qualitative vs. quantitative.*

[AR]: We changed the wording to "satellite-derived atmospheric composition" in this sentence. In Section 5.5.1 of the original manuscript, we loosely discuss how the observations should be used (e.g. "By comparing the results of experiments 1, 2, 3, and 6 to the observations discussed in Section 4.4 (and listed in Table S1), specific model inputs and processes can be evaluated"). The detailed quantitative and qualitative model evaluation can, and most likely will be its own paper(s), as those studies are carried out. To include much more on that topic would be beyond the scope of this paper, which is more focused on the motivation and model design (and already quite long). Our paper does gather up from the community a list of the relevant observations that could be used for model evaluation, but is not overly prescriptive on how that gets carried out.

*Lines 668 – 669: Please provide a list of LIDAR stations and available measurements time periods that will be used in this study.*

[AR]: In the revised manuscript, we have stated more generally that regular monitoring measurements of the pollutants can be used for model evaluation, as to list them all from all jurisdictions globally would be a monumental task. We have opted to only keep the relevant measurement campaign data in Table S1, and we have removed the sentence about LIDAR observations.

*Lines 669 – 671: "All surface monitoring measurements of the pollutants in Section 3.1 could be used for model evaluation" – it is unclear what "could" means in this context. Will HTAP3 Fires model evaluation activities make use of all such measurements? If so, "are expected" instead of "could" might be a better wording. Also, why does Table S1 list BC and CO surface monitoring data from EMEP over Europe but not corresponding data from NAPS and AQS over Canada and the US? AERONET data also likely would be very useful for this model evaluation activity.*

[AR]: Thank you for pointing out this inconsistency in measurement datasets and wording in this section. We have changed all wording to "can" or "could" be used for model evaluation, and the original text in Section 4.4 mentions that atmospheric monitoring networks from all jurisdictions are available for model evaluation. In the revised supplement, we removed those regular monitoring datasets (e,g, EMEP) from Table S1 for consistency and because to include *all* monitoring datasets for all countries globally would be too big a job for this model-focused paper. We instead focus this table on lesser-known and more fire-relevant observations, such as those from field campaigns.

*Line 671: change "suggestion" to "suggested"*
[AR]: Thank you. This has been corrected.

*Lines 681 – 684: The wording "can be used" leaves it unclear whether such cross-disciplinary satellite and in-situ data deposition analyses will be performed as part of HTAP3-Fires. If there are such plans for evaluating deposition, more details are needed on which types of models and modeling periods such an approach would be most applicable for and how it will be implemented.*

[AR]: The satellite-derived measurements of deposition can be used to evaluate all models' results, since all model types described in this paper include deposition of pollutants in their atmospheric processes. As noted in one of our responses above, we aim to balance the model scope, design options, and guidance in this paper. We do not include a lot of additional details for model evaluation, as this would lengthen the paper further. We instead aimed to highlight some opportunities so that participants are aware and can place their results in the context of available observations.

*Lines 685 – 690: This paragraph does not seem to directly relate to how observational data will be used for model evaluation.*

[AR]: Thank you. We have moved this paragraph to Section 3.1.6 in the revised manuscript.

*Lines 702 – 703: Preferably, an experimental design description paper like this one would have such key decisions already settled when written and submitted. Given that this sentence suggests that this aspect is still in flux, more details are needed on which criteria will be used to determine that reliable emission assessments are available that support the selection of specific time period(s). Time period(s) when relevant observations are available should already be known.*

[AR]: The time periods are known and given in Section 5.1. However, we can see how this section, where we discuss the options considered/how the decision was made, makes it sound like we haven't decided yet. We have revised the manuscript to (a) provide more context on the purpose of Section 4 at the end of the introduction, and (b) in this section particularly, we change the wording to be more certain: e.g. "These time periods are selected based on…"

*Lines 705 – 710: When will this identification of short-term case studies to be analyzed in HTAP3 Fires be made? What is the process for selecting specific case studies?*

[AR]: All of these three case studies mentioned are suggested, given that there are either many measurements or because they were extreme events. They are suggested for regional models who are focused on those domains and would prefer to just run one year or less. However, at this time, we did not ask for a 100% firm commitment from modelling centres, so it remains to be seen which model(s) will run which case study. The model design details for the case studies can be found throughout section 5 (e.g., in Tables 3 and 5, and in section 5.5.1.

*Lines 711 – 736. The definition of regions to be used for the perturbation experiments is critical to the design of the modeling study described in this paper. This section provides an unclear message about whether the process for defining these regions is still ongoing (e.g. "these [HTAP2] source regions should be further defined", "proposed merged regions") or whether the 8 merged GFED regions shown in Figure 3d already reflect the final definition of regions that will be used. Preferably, it is the latter, in which case the wording in the paragraph should be revised to reflect that Figures 3a – 3c present different potential starting points while Figure 3d shows the final decision, arrived at after making the considerations described in this paragraph. If the region definition process is still in progress, this needs to be clearly stated and would be a fairly major limitation of publishing this experimental design paper now rather than later when such a key decision has been finalized.*

[AR]: Thank you for this note. Yes, Fig 3d is the final decision, and we have revised Section 4.5.2 (and the Figure 3 caption) to be more clear that Figures 3a-c were all considered and discussed as starting points only. Note that as mentioned above, Section 4 was to discuss the options and Section 5 to provide the decisions (in this case, it was given in Section 5.3, Table 5, Exp #5). Hopefully the revised manuscript as a whole makes this more clear as well.

*Line 758 - 759: The different parts of this sentence don't quite mesh together, consider revising maybe along these lines: "Much of the uncertainty in the wider impacts of fires arises from weakness in our understanding of fire processes, their representation in models, and the sensitivity of the impacts to these treatments"*

[AR]: Thank you for pointing this out. As reviewer #1 also had a similar comment, we have revised this first paragraph of Section 4.5.3 entirely to be clearer.

*Line 768: Rather than being "less efficient" in the free troposphere than the boundary layer, dry deposition only occurs at the surface, not the free troposphere.*

[AR]: Thank you for this note. While dry deposition only occurs at the surface, wet deposition starts higher in the atmosphere with wet scavenging. We have removed "wet and dry" from the text in the revised manuscript to correct the text.

*Line 774: remove comma after "whereas"*

[AR]: Removed in the revised manuscript.

*Lines 774 – 776: "Daily information on wildfire injection heights … can be used in the calculation of injection heights". This is unclear – if daily information on injection heights is available, why do they need to be calculated?*

[AR]: Thank you for catching this. We have revised to "can be used in the calculation of the vertical distribution of fire plumes", as models vary in how they distribute the fire emissions vertically based on a given injection height.

*Line 780: change "these models" to "the models"*

[AR]: Done.

*Line 789: Please specify what type of data from these platforms would be used for evaluating the effects of plume rise, whether such evaluation would be quantitative or qualitative in nature, and which time and space scales it would be performed for.*

[AR]: We have revised the text to clarify that we meant the fire plume heights that are derived from some satellite measurements, as well as the vertical distribution of fire pollutants from aircraft measurements, and that this model evaluation would be a quantitative comparison. With satellite data, this can be done globally, but with aircraft data, regionally.

*Lines 802 – 808: building upon the third factor listed here, the study might also want to specifically explore how the use of different chemical mechanisms and aerosol schemes affects simulated impacts, holding all other aspects constant.*

[AR]: That would be an interesting study, however, within one model, it is not always the case that more than one chemical mechanism or more than one aerosol scheme is installed/implemented (and is therefore not necessarily an intercomparison topic that our study is focused on). It would be valuable to explore how chemical complexity in VOC schemes influenced results and we have added this suggestion in Section 5.3.1 of the manuscript.

*Line 823: remove "to" before "turn"*

[AR]: Done.

*Line 830: suggest changing the first occurrence of "provided" to "if" (or "when", as applicable)*

[AR]: Thank you for pointing out this awkward sentence. We have revised to remove the repetition and to be more clear "Future modelling experiments can be performed with chemical transport models that use provided future emissions and meteorology (see Section 5.2)."

*Line 842: remove comma after scales*

[AR]: Done.

*Line 853, Table 3: Should the future medium option list 2010 – 2020 and 2045 – 2055, instead of 2015 – 2050? The discussion in the text mentions two ten year periods*

[AR]: Yes, we have revised Table 3 to contain this as the medium option (2010-2020 and 2040-2050), and the long option to run all the way through (2010-2050).

*Line 873: please update the timeline for the release of these emissions*

[AR]: HTAPv3.1 and GAINS LRTAP are both expected in January 2025 (imminently). We have updated the text to indicate that they are available and now have a complete Data Availability section with links and references of the revised manuscript.

*Table 4: The biogenic and other natural emissions section list "MEGAN or models' own", but the text says "we suggest that each modeling centre use their preferred emissions from biogenic and other natural sources". This makes it unclear whether a reference MEGAN dataset (driven by which reanalysis / future meteorology?) will be provided to groups who would like to use it, or if there is no common fallback dataset.*

[AR]: Thank you for pointing out this inconsistency. While we did mention in Section 4.2.4 that the majority of models use MEGAN, we will not be prescribing biogenic emissions for this modeling study, which is why the text of Section 5.2 did not mention MEGAN. We removed MEGAN from Table 4 in the revised manuscript.

*Table 4: The notes/references under historical fire emissions state "Note: these will be updated in the near future to include newer emission factors". Has this happened already? If not, when will it happen, and are groups expected to use the files with the newer emission factors?*

[AR]: Yes, apologies. In the time since we originally submitted this paper (~6 months ago), this was done and the emissions now available. All text related to the historical BB emissions dataset has been updated in the revised manuscript.

*Table 4: The timeline for the historical anthropogenic emissions in the notes/references section is outdated, please update. Please also change "TBD" to the actual download location, it is critical that this information is available before the paper is published.*

[AR]: Yes, apologies. In the time since we originally submitted this paper (~6 months ago), this was done and the emissions are now either available, or imminently available (so will be able to provide the download link in the Data Availability Section before final publication). All text related to the historical anthropogenic emissions dataset has been updated in the revised manuscript.

*Section 890 – 894: see my earlier comments about also discussing models that use meteorology by models like WRF, nudged towards reanalysis fields. What is the recommended protocol for these types of models?*

[AR]: Thank you for this note. The suggestion that modellers use "their preferred meteorology" stands for those types of models as well.

*Section 5.3 / Table 5 / description of experiments: more detail is needed about the plans for exp2 (case studies) and exp8 (data assimilation)*

[AR]: In the revised manuscript, we have added more detail for these two experiments in Table 5 and Section 5.3. E.g., there's a new section 5.3.3 on the data assimilation experiments, suggesting the assimilation of MOPITT and OMI data to constrain CO, O3, and NO2 BB emissions from GFAS4HTAP. We also now provide the region and year for the case studies in Table 5. And please note that a detailed technical model guidance document will be circulated to participants in the near future covering some detailed technical guidance that is not necessarily required for this publication, but will be useful to clarify details for modellers.

*Section 5.3.1, lines 911 – 919: I suggest also considering perturbations to VOC speciation and the volatility distribution of total emitted reactive organic carbon*

[AR]: We have added the following to the revised manuscript "For models with suitable capability, exploration of the effects of different levels of complexity in VOC chemistry or differences in volatility or reactivity of VOC".

*Lines 917 – 919: what is the "base" temporal resolution of fire emissions in exp1 – hourly, daily?*

[AR]: While some CTMs and most ESMs typically use monthly fire emissions by default, we do not explicitly recommend monthly for baseline experiments since we know (as per section

4.2.2) that higher temporal resolution for fire emissions is very important for better simulations of BB and its impacts. Therefore, the guidance for baseline experiments is for models to use their own preferred temporal resolution for fire emissions input.
This has been clarified in the revised manuscript (Section 5.2 and guidance document).

*Line 946: This should be Table S1, not S0. Also see my earlier comment regarding the availability of CO, BC, and a range of other gaseous and aerosol pollutants from NAPS and AQS. How will the AGES dataset be used since 2023 is not included in the medium or long simulation period options? Will modeling 2023 be required for models conducting "short" case study simulations?*

[AR]: Table reference corrected to S1. Recent time periods like 2023 are difficult to simulate accurately since historic anthropogenic emissions datasets are not yet available for that year (e.g. HTAP v3.1 only goes up to 2020). It is also not one of the 3 time periods we ultimately suggested for the case study experiments. That said, the purpose of Table S1 was twofold: (a) to suggest observational datasets that would be helpful in model evaluation, and (b) help in the decision-making process for the time periods to simulate (with decision being given in Table 3). The AGES dataset and many others in this table were included for that second purpose. However, in the revised manuscript (Section 4.4 and 5.5.1, we clarify that not all of those measurements may be used. Just those overlapping in time and location can be used, as well as others not included in Table S1.

*Lines 945 – 954: Will the datasets listed in Table S1 be curated by HTAP3 Fires to provide a common set of variable names, units, metadata, and method and sampling interval information to participants performing model evaluation, or will participants be expected to obtain the raw data from all the different sources listed in Table S1 themselves and then prepare them for model evaluation?*

[AR]: No, while that would make model evaluation infinitely easier, at this time, no one from this project has committed to do that. Doing so may also conflict with the Intellectual Property and distribution policies of some of the measurement datasets.

*Are the community tools listed in this section set up to easily ingest all of the different observational datasets listed in Table S1?*

[AR]: No, they are just suggestions for the model evaluation phase that has not yet begun.

*More generally, model evaluation is a key component of this activity, and the authors are correct in stating that it will require a large effort by the community. It would be good if this paper could provide a clearer roadmap for how this critical model evaluation task will be accomplished as part of HTAP3 Fires, including the creation of infrastructure for obtaining and harmonizing observations and outlining how meaningful evaluation efforts will be structured given the diversity of models and temporal and spatial scales of the expected model outputs. This roadmap should also include discussion whether the HTAP3 Fires organizers expect to provide leadership on model evaluation or whether this will entirely rely on community volunteers.*

[AR]: Analysis, including model evaluation, will be driven by the interests of different groups, and we are not able to anticipate all possible uses here. As mentioned in addressing a

previous comment, what you describe could be a whole paper on its own. It is beyond the scope of the current paper to elaborate that much further about the model evaluation stage, which will be driven entirely by community volunteers. We do however plan to provide all the model output generated in this project in a consistent file format and hosted in one openly-accessible repository. Those details will be included in the technical guidance document that will be circulated in the near future to participants when the model simulations are launched.

*Lines 964-965: please define "high spatial and temporal resolution" and discuss what resolution is needed for such novel health risk assessments, as this will inform the required design of model simulations. Please also discuss how these anticipated health risk assessments would account for model biases, e.g. by employing data fusion or other bias correction techniques, and provide details on such planned analyses.*

[AR]: Note that we corrected the table references in Section 5.5.2 of the revised manuscript. Spatial and temporal resolution varies for the models participating in this study and we cannot provide a single value. However, in the revised manuscript, we provided the orders of magnitude for these values to guide the reader. We also thank the reviewer for commenting on using data fusion / correction techniques which are widely used in health impact assessments. Hence we have amended the text as follows:
"The surface-level model outputs of atmospheric composition at high spatial (O(10 km) for global, O(1km) for regional models) and temporal (monthly down to daily) resolution will be invaluable for new health risk assessments, especially when fused with other modelling (e.g., land-use regression) and observational (e.g., remote sensing) techniques (Johnson et al., 2020)."

*Lines 966 – 971, section 5.5.3: This section lacks specificity of the planned analyses.*

[AR]: We have added more details in the revised manuscript. The revised paragraph reads:
"Climate impacts can be assessed through the RF from fire-emitted pollutants by comparing the differences of the radiative fluxes of the simulations with and without fire emissions (i.e. the Baseline simulation and the regional and sectoral emissions perturbations). To assess the component specific RFs more detailed simulations with source attribution techniques for example for O3 (Grewe et al. (2017) and Butler et al. (2018)), or for aerosols (Righi et al. (2021)) are helpful. Models capable of such possibilities therefore perform additional pollutant specific perturbations including source attribution techniques. Moreover, the model's composition fields can be applied in offline radiative transfer models or via the kernel method to calculate the component specific RF."